

# Long-term characterisation of the vertical structure of Saharan dust outbreaks over the Canary Islands using lidar and radiosondes profiles: implications for radiative and cloud processes over the subtropical Atlantic Ocean.

África Barreto[1,2], Emilio Cuevas[1], Rosa D. García[3,1], Judit Carrillo[4], Joseph M. Prospero[5], Luka Ilić[6], Sara Basart[7], Alberto J. Berjón[3,1], Carlos L. Marrero[1], Yballa Hernández[1,8], Juan J. Bustos[1], Slobodan Ničković[6], and Margarita Yela[9]

[1]Izaña Atmospheric Research Center (IARC), Agencia Estatal de Meteorología (AEMET), Spain
[2]Atmospheric Optics Group of Valladolid University (GOA–UVa), Valladolid University, Valladolid, Spain
[3]TRAGSATEC, Madrid, Spain
[4]Departamento de Física, Universidad de La Laguna (ULL), Canary Islands, Spain
[5]Rosenstiel School of Marine and Atmospheric Science, University of Miami, Miami, USA
[6]Republic Hydrometeorological Service of Serbia, Belgrade, Serbia
[7]Barcelona Supercomputing Centre, Barcelona, Spain
[8]Consejería de Educación, Universidades y Sostenibilidad, Gobierno de Canarias, Canary Islands, Spain
[9]Instrumentation and Atmospheric Research Department, National Institute for Aerospace Technology (INTA), Madrid, Spain

**Correspondence:** A. Barreto (abarretov@aemet.es)

**Abstract.** Every year, large-scale African dust outbreaks frequently pass over the Canary Islands (Spain). Here we describe the seasonal evolution of atmospheric aerosol extinction and meteorological vertical profiles at Tenerife over the period 2007 – 2018 using long-term Micropulse Lidar (MPL-3) and radiosondes observations. These measurements are used to categorise the different patterns of dust transport over the subtropical North Atlantic and, for the first time, to robustly describe the dust vertical distribution in the Saharan Air Layer (SAL) over this region. Three atmospheric scenarios dominate the aerosol climatology: dust-free (clean) conditions, the summer-Saharan scenario (Summer-SAL) and the winter-Saharan scenario (Winter-SAL).

A relatively well-mixed marine boundary layer (MBL) was observed in the case of clean (dust-free) conditions; it was associated with lidar extinction coefficients $(\alpha) \sim 0.030$ km$^{-1}$ with minimum $\alpha$ $(< 0.022$ km$^{-1})$ in the free troposphere (FT). The Summer-SAL has been characterised as a dust-laden layer strongly affecting both the MBL ($\Delta\alpha = 48$ % relative to clean conditions) and the free troposphere. The Summer-SAL appears as a well-stratified layer, relatively dry at lower levels but more humid at higher levels compared with clean FT (CFT) conditions ($\Delta r \sim$ -44 % at the SAL's base and $\Delta r \sim$ +332 % at 5.3 km, where $r$ is the water vapour mixing ratio), with a peak of $\alpha > 0.066$ km$^{-1}$ at $\sim$ 2.5 km. Desert dust is present up to $\sim$ 6.0 km, the SAL top based on the altitude of SAL's temperature inversion (STI). In the Winter-SAL scenario, the dust layer is confined to lower levels, below 2 km altitude. This layer is characterised by a dry anomaly at lower levels ($\Delta r \sim$ -38 % in comparison to the clean scenario) and a dust peak at $\sim$ 1.3 km height. CFT conditions were found above 2.3 km.

Our results reveal the important role that both dust and water vapour play in the radiative balance within the Summer- and Winter-SAL. The dominant dust-induced shortwave (SW) radiative warming in summer (heating rates up to +0.7 K day$^{-1}$)



is found slightly below the dust maximum. However, the dominant contribution of water vapour was observed as a net SW warming observed within the SAL (from 2.1 km to 5.7 km) and as a strong cold anomaly near the SAL's top (-0.6 K day$^{-1}$).

The higher water vapour content found to be carried on the Summer-SAL, despite being very low, represents a high relative variation in comparison to the very dry clean free troposphere in the subtropics. This relevant aspect should be properly taken into account in atmospheric modelling processes. In the case of the Winter-SAL, we observed a dust-induced radiative effect dominated by SW heating (maximum heating of +0.7 K day$^{-1}$ at 1.5 km, near the dust peak); both dust and atmospheric water vapour impact heating in the atmospheric column. This is the case of the SW heating within the SAL (maximum near the $r$

peak), the dry anomaly at lower levels ($\Delta r \sim$ -38 % at 1 km) and the thermal cooling ($\sim$ 0.3 K day$^{-1}$) from the STI upwards.

Finally, we hypothesise that the SAL can impact heterogeneous ice nucleation processes through the frequent occurrence of mid-level clouds observed near the SAL top at relatively warm temperatures. A dust event that affected Tenerife in August 2015 is simulated using the regional DREAM model to assess the role of dust and water vapour carried within SAL in the ice nucleation processes. The modelling results reproduce the arrival of the dust plume and its extension over the island and

confirm the observed relationship between the Summer-SAL conditions and the formation of mid- and high-level clouds.

## 1 INTRODUCTION

Dust is one of the main components of the atmospheric aerosol load, representing about 75 % of the global aerosols injected into the atmosphere (Kinne et al., 2006; Huneeus et al., 2011; Mona et al., 2012; Wu et al., 2020). Mineral dust emitted from the Sahara and surrounding regions contributes to more than half of the global dust emissions (IPCC, 2013).

The African dust is transported westwards as a hot, dry, dust-laden elevated layer (the so-called Saharan Air Layer or SAL) over the North Atlantic for thousands of kilometres (Carlson and Prospero, 1972; Prospero and Carlson, 1972; Karyampudi and Carlson, 1988; Karyampudi et al., 1999; Engelstaedter et al., 2006; Sunnu et al., 2008; Rodríguez et al., 2011). These large amounts of Saharan mineral dust within the SAL are advected off the African continent to the Caribbean (Prospero and Carlson, 1972; Carlson and Prospero, 1972; Prospero and Mayol-Bracero, 2013; Prospero et al., 2021), the Amazon rainforest (Yu et al.,

2015; Prospero et al., 2020) or the Mediterranean basin (Carlson and Prospero, 1972; Hamonou et al., 1999; Reid et al., 2003). A large number of studies in the literature have found that mineral dust causes a strong radiative impact on the atmospheric vertical stability, thermodynamics and wind shear (Carlson and Benjamin, 1980; Karyampudi et al., 1999; Quijano et al., 2000; Karyampudi and Pierce, 2002; Dunion and Velden, 2004; Kim et al., 2004; Wong and Dessler, 2005; Satheesh et al., 2007; Zhu et al., 2007; Chen et al., 2010, among others). Recently, some authors have shown that, although mineral dust produces net

radiative heating within the SAL in summer, the small amount of water vapour that this layer contains, significantly higher than that present in the clean free troposphere, represents the major radiative driver for the dust layer in summer (Gutleben et al., 2019, 2020). Furthermore, many studies in the literature confirm that mineral dust within the SAL also plays an important role as cloud condensation nuclei (CCN), giant CCN (GCCN) and ice nuclei, affecting the properties of clouds and their indirect effect on radiative balance and consequently on climate (Sassen et al., 2003; DeMott et al., 2003; Murray et al., 2012; Boose

et al., 2016).



Most of the studies on the SAL have been focused on dust transport to the Caribbean region through the northern equatorial Atlantic. Between the tropics and the Canary Islands, there is no place where the SAL can be analysed and characterised once it has left the African continent and has docked on the subtropical North Atlantic marine boundary layer (MBL), impacting the subtropical free troposphere (FT). Another problematic point of the current studies is related to the preferential study of the dust leaving Africa at tropical latitudes in the summer season when the Saharan outbreaks over the North Atlantic are mostly confined in an elevated mixed layer (Carlson, 2016; Prospero and Carlson, 1980). As Tsamalis et al. (2013) showed, the SAL can be identified near the west coast of Africa all year round with a clear spatial seasonal cycle: between 5°S and 15°N in winter, and 5°–30°N in summer. Therefore, the Canary Islands lie on the northern edge of the dust belt in summer and are significantly affected by the SAL. The study of the SAL at subtropical latitudes will add important additional information since this location is representative of the almost pure Saharan convective boundary layer (CBL).

The transport of dust in the subtropics presents a stronger seasonal dependence than at tropical latitudes, with a marked maximum in summer. In this season, the different mesoscale mechanisms triggering the activation of some of the multiple Saharan dust sources are modulated by longitudinal shifts of the Saharan Heat Low (SHL) (Lavaysse et al., 2009), which are closely related to changes in the North African Dipole Intensity (NAFDI) (Rodríguez et al., 2015; Cuevas et al., 2017a). In tropical latitudes, the SAL observed in summer over the ocean is formed largely as a consequence of the dust transport from the Sahara, and therefore the atmospheric processes that explain this dust-laden layer are similar to those of the subtropical SAL. However, the development of mesoscale convective systems during the wet season driven by the Monsoon and the Inter-Tropical Convergence Zone (ITCZ) activates additional dust source at tropical latitudes (Marticorena et al., 2010; Tegen et al., 2013).

In winter, most of the dust in the eastern tropical North Atlantic is transported within the marine layer in a shallowest layer, up to 1.5 - 3 km (Prospero and Carlson, 1980; Chiapello et al., 1995; Zhu et al., 2007; Ansmann et al., 2011; Tsamalis et al., 2013; Rittmeister et al., 2017; Weinzierl et al., 2017; Senghor et al., 2017). Winter dust mobilization in the Sahel region is often associated with the Harmattan winds that transport dust southwestward from the Bodélé and adjacent areas (Marticorena et al., 2010) at low levels (Cavalieri et al., 2010). During this dry season, dust is frequently mixed with biomass-burning aerosols confined in the upper layers (Haywood et al., 2008; Cavalieri et al., 2010). In contrast, the atmospheric mechanisms that cause dust intrusions over the subtropical North Atlantic in winter are generally linked to baroclinic processes and are usually of short duration and limited geographic extension. These baroclinic processes are often located in the vicinity of the Canary Islands and affect Western Sahara, Northern Mauritania and Western Algeria. Therefore, this dust is free of biomass-burning aerosols. An outstanding example was the huge dust intrusion that occurred in February 2020 (Cuevas et al., 2021). Although it is not strictly accurate to talk about the SAL in winter since the intrusions on the ocean do not constitute an elevated layer, in this study we will refer to them as "Winter-SAL". The impact of the Winter-SAL on the subtropical MBL, and more specifically on the Canary Islands, was analysed by Alonso-Pérez et al. (2007, 2011, 2012).

The altitude to which the aerosols are initially lifted and in which they are subsequently transported will influence their lifetime and radiative effects (Generoso et al., 2008). Consequently, there is a need for systematic ground-based mineral dust





and vertical thermodynamic observations near dust source regions to understand the evolution of the 3D-spatial and temporal dust distribution and the role played by desert dust in the effective radiative forcing (ERF).

Lidars offer the most efficient technique for the identification and characterisation of the vertical distribution of atmospheric aerosols. However, the quantitative data inversion in the case of elastic backscatter lidars involves knowledge of the relationship between aerosol backscatter ($\beta$) and extinction coefficients ($\alpha$), that is, the extinction-to-backscatter ratio ($LR = \alpha/\beta$ or lidar

ratio) (Fernald, 1984; Klett, 1985). This presupposes knowledge of the vertical aerosol distribution, critical in the case of complex aerosol distributions such as in elevated layers. The uncertainty on this parameter determines the accuracy of the retrieved profiles (Cattrall et al., 2005). The accuracy of the retrievals can be improved by combining active and passive remote sensing aerosol observations. The synergetic aerosol monitoring of co-located photometer and lidar observations is a powerful tool for aerosol research (Ansmann et al., 2012; Lopatin et al., 2013; Mortier et al., 2013; Binietoglou et al., 2015; Chaikovsky

et al., 2016; Bovchaliuk et al., 2016; Berjón et al., 2019).

In this study, we describe the seasonal evolution of the vertical atmospheric profiles over the subtropical North Atlantic region on Tenerife, Canary Islands, using 12 years of observations (2007- 2018) with a Micropulse Lidar (MPL-3). In addition, we use radiosonde measurements from Tenerife. We use concurrent sun photometer measures to minimize the uncertainties involved in the aerosol extinction retrieval from an elastic backscatter system. To assess the radiative effects, we use the two-

layer approach presented in Berjón et al. (2019) in which two-column lidar ratios ($LR_1$ and $LR_2$) are calculated as inputs, retrieved from the Aerosol Optical Depth (AOD) information in two layers (marine layer and free troposphere). This technique yields a better description of the $LR_i$ than the standard one-layer approach. Sect. 2 describes the test site. All the instruments and ancillary information are described in Sect. 3. In Sects. 4.1 and 4.2 we characterise the lower troposphere under clean (dust-free) conditions (in four different seasons covering the whole year) and under Saharan-dust conditions (focused on winter and

summer seasons). Sects. 5 and 6 present the main results on the SAL's impact on the atmospheric vertical thermodynamic aerosol extinction profiles as well as the corresponding radiative impact of this dusty layer. Sect. 7 studies the potential role of the SAL in summer and winter on the formation of mid-level clouds as a result of the activation of heterogeneous ice nucleation processes. Finally, Sect. 8 presents the summary and the main conclusions from this work.

## 2 EXPERIMENT SITE

The Canary Islands, located in the subtropical North Atlantic, are characterised by a very stable and well stratified lower troposphere with strong temperature inversions (Cuevas, 1995; Dorta, 1996; Carrillo et al., 2016). These are modulated by quasi-permanent subsidence conditions, as a result of the descending branch of the Hadley-cell. Carrillo et al. (2016) made a comprehensive analysis of the temperature inversions in the lower troposphere of the Macaronesian region, including the Canary Islands. They found the presence of one or two sharp temperature inversions or transition levels below the 700 hPa

level. The first transition level in the double-inversion structure coincides with the maximum extent of the surface-related turbulent mixing which limits the dry adiabatic convection and caps the humid and well-mixed MBL. This transition level, denoted as the marine boundary inversion (MBI), is characterised by a strong vertical humidity gradient. The second transition





level, denoted as the trade wind inversion (TWI), is located at the top of a highly stable layer, the trade wind layer (TWL), normally associated with synoptic-scale subsidence processes which limit vertically the quasi-permanent area of stratocumulus clouds frequently observed at this latitude. According to these authors, the percentage of occurrence of this double structure (normally found between 900 and 800 hPa) is low, with a maximum occurrence of 33 % in summer. Consequently, in this study we use a two-layer approach (one single temperature inversion) where the MBI and the TWI are the same temperature inversion capping the MBL, and separating it from the FT above. It is important to note that the study by Carrillo et al. (2016) on temperature inversions was carried out under no-dust conditions. However, the characteristics of the temperature inversions are modified by the impact of the SAL which introduces a new temperature inversion in the lower troposphere. These aspects are also addressed in our study.

## 3 INSTRUMENTS AND ANCILLARY INFORMATION

This study focuses on the observations and modelling results obtained over two locations in Tenerife: Santa Cruz de Tenerife station SCO ($28.5°$N; $16.2°$W at 52 m a.s.l.) and the high-altitude Izaña Global Atmospheric Watch (GAW) Observatory IZO ($28.3°$N; $16.5°$W at 2391 m a.s.l), horizontally spaced less than 30 km. These two sites belong to the State Meteorological Agency (AEMET) and are managed by the Izaña Atmospheric Research Centre (IARC) (Cuevas et al., 2017b).

### 3.1 AOD from AERONET

Column-integrated aerosol optical depth (AOD) and Ångström Exponent (AE) data have been collected at IZO and SCO AERONET sites. AERONET (Aerosol Robotic Network, http://aeronet.gsfc.nasa.gov, last access: 1 March 2021) is a globally-distributed federated network for aerosol optical properties using the CE-318 sunphotometer as a standard instrument (Holben et al., 1998; Giles et al., 2019). Ground-based CE-318 sun measurements were performed at eight nominal wavelengths (340 to 1020 nm) with an approximate field-of-view of $1.3°$ (Holben et al., 1998; Torres et al., 2013). AERONET's AOD uncertainty is spectrally dependent; is 0.01 at 500 nm, the approximate spectral wavelength used in the present study. Following Eck et al. (1999); Schuster et al. (2006), these uncertainties alter the AE by 0.03-0.04. We used cloud screened and quality assured AOD data from AERONET level 2.0 (Smirnov et al., 2000).

### 3.2 Aerosol extinction from MPL-3

The micropulse backscatter lidar version 3 (MPL-3), extensively described in Spinhirne et al. (1995); Campbell et al. (2002), contains a solid-state Nd:YLF laser system which emits at 523 nm. It has a high-pulse repetition rate of 2500 Hz and eye-safe energy levels ($\sim 7 \ \mu$J) by expanding the low-power transmitted beam using a Cassegrain telescope. The same configuration is used for transmission of the energy pulse and detection of the backscattered light. Range resolution is 75 m. Operationally, it can detect returns at distances up to 20 km. Efficient detection is achieved by an avalanche photodiode. The MPL is an autonomous instrument, operational in full-time continuous mode (24h, 365 days/year). This instrument is the reference instrument of the NASA Micro-Pulse Lidar Network (MPLNET) (Welton et al., 2005), a federated global network contributing network to the





World Meteorological Organization (WMO) GAW Aerosol Lidar Observation Network, GALION. The MPL was developed
to obtain measurements of cloud scattering cross-sections and optical thickness, planetary boundary layer height, aerosols
extinction and optical thickness profiles (Campbell et al., 2002). The resulting signal extracted from the raw MPL data after
the correction process is applied is called the normalized relative backscatter (NRB) signal (Welton and Campbell, 2002). This
requires the application of background, dead time, afterpulse and overlap corrections (Campbell et al., 2002). Furthermore, the
corrected profiles are hourly averaged to increase the signal-to-noise ratio (SNR). The MPL-3 installed at SCO station achieves
a full overlap at an average of $\sim 5$ km. Welton and Campbell (2002) estimated a relative uncertainty in the overlap correction
between 5 % and 10 %, and is most important in the lowermost atmospheric layers. Due to these large uncertainties for an
incomplete overlap in addition to the saturation of the detector in the near range caused by the afterpulse phenomenon, the
information below 300 m is disregarded.

We have used the lidar-sunphotometer synergy to reduce the uncertainty in $LR$ and thereby to improve the aerosol charac-
terisation (Ansmann et al., 2012; Lopatin et al., 2013; Binietoglou et al., 2015; Chaikovsky et al., 2016; Berjón et al., 2019).
AODs (500 nm) at SCO and at IZO AERONET stations (see Section 3.1) have served as a constraint to fix more accurately
$LR$ in the two-layer approach, already presented by Berjón et al. (2019), and therefore retrieving a more accurate value for
the aerosol extinction coefficient ($\alpha$) at 523 nm. This iterative procedure enables the $LR$ to be calculated for each layer, the
boundary layer (principally containing marine and dust aerosols) and the lofted aerosols in the free troposphere (mainly dust
in both clean and Saharan conditions). Since $LR$ is an intensive aerosol property that does not depend on the number density
of the aerosol, the two-layer approximation allows a quantitative description of clean conditions as well as the situations with
complex aerosol layering typically related to Saharan dust outbreaks.

Following Berjón et al. (2019), cloud-free conditions in the one- and two-layer approach are ensured by means of the
AERONET cloud-screened product as the first filter, and also using a specific cloud-screening for lidar data based on the
smoothness of the lidar background (Clothiaux et al., 1998). Therefore, the two-layer procedure is more restrictive than the
single method itself since cloudless conditions and output AOD control are simultaneously required for both stations (Berjón
et al., 2019). See Berjón et al. (2019) for further details on the two-layer procedure.

### 3.3    Meteorological vertical profiles from radiosondes

Meteorological vertical profiles are provided by radiosondes launched at 12 UTC in 2007-2018 at AEMET's Güímar station
(28.3°N; 16.4°W at 105 m a.s.l.; WMO GUAN #60018), located at the coastline, approximately 22 km from SCO and 11 km
from IZO. Temperature ($T$), relative humidity ($RH$) and pressure ($P$) are measured using Vaisala RS92 radiosondes, while a
Global Positioning System (GPS) windfinding is used for wind speed and wind direction calculation (Rodríguez-Franco and
Cuevas, 2013; Carrillo et al., 2016). Potential temperature ($\theta$), equivalent potential temperature ($\theta_e$) and water vapour mixing
ratio ($r$) are computed from the basic meteorological sonde parameters.



### 3.4 FLEXTRA back-trajectories and the African air-masses Residence Time Index (ARTI)

The African air masses Residence Time Index (ARTI) is an objective index developed in this study to discriminate between Saharan dust-laden air masses and those from clean or non-dust air masses. ARTI corresponds to the percentage of the time in which the air-mass trajectory is within the Saharan-Sahel area. For the ARTI calculations, the FLEXible TRAjectories model (FLEXTRA; https://folk.nilu.no/ andreas/flextra.html) (Stohl et al., 1995; Stohl and Seibert, 1998) is used for the determination of the origin of the different air masses arriving at Tenerife. FLEXTRA is a kinematic trajectory model developed at the Institute of Meteorology and Geophysics in Vienna that uses input data from the European Centre for Medium-Range Weather Forecasts (ECMWF). FLEXTRA was run to provide 120-hour back-trajectories arriving at two levels, 150 m and 2400 m, the most representative levels to study the long transport arriving at SCO and IZO, respectively. Saharan dust-laden air masses (ARTI > 0) can be discriminated from other clean or non-dust air masses (ARTI = 0), such as those associated with the trade wind regime within the MBL or airflows from northernmost latitudes and upper-middle troposphere over the North Atlantic in the FT. This restriction in the ARTI for clean conditions, although very restrictive, ensures that we correctly select clean scenarios for its characterisation.

### 3.5 Radiative Transference simulations

Atmospheric heating rate calculations are performed using the LibRadtran radiative transfer model (Mayer and Kylling, 2005; Emde et al., 2016). The simulations are done with the radiative transfer solver DISORT (DIScrete Ordinate Radiative Transfer) (Stamnes et al., 1988) for spectral ranges of 280 to 4000 nm (shortwave; SW) and 4000 nm and 100 $\mu$m (longwave; LW). The absorption parameterization REPTRAN (Gasteiger et al., 2014) included in the LibRadtran is used in both spectral ranges. Regarding the atmosphere gas composition, we use the standard midlatitude summer atmosphere model (Anderson et al., 1986). The temperature and relative humidity profiles are taken from the monthly or seasonally median profiles obtained from radiosondes (Sect. 3.3).

The key input parameters required for calculating the radiative heating rates are AOD (or $\alpha$), single scattering albedo ($SSA$) and asymmetry parameter ($g$), which are all wavelength dependent. MOPSMAP (Modeled optical properties of ensembles of aerosol particles) (Gasteiger and Wiegner, 2018) (https://mopsmap.net, last access: 1 March 2021) is used to reconstruct the required input aerosol parameters under different atmospheric conditions. A similar approach is used in different fields by (Barreto et al., 2020; Piontek et al., 2021; Chen-Chen et al., 2021; Jiang et al., 2021). This package consists of a dataset of pre-calculated optical properties of different user-defined aerosol ensembles under a high variety of atmospheric conditions. One of the most important advantages of this database in comparison to the Optical Properties of Aerosols and Clouds (OPAC) (Hess et al., 1998) database is that MOPSMAP includes more aerosol shape features such as spheroids and irregularly shaped dust particles. The details of the different optical properties extracted from MOPSMAP in the current study are given in Sect. 6.





### 3.6 Ice Nucleating Particle Parameterizations in Dust Regional Atmospheric Model (DREAM)

We obtain predictions of ice nucleation particle concentration (INPC) using the Dust Regional Atmospheric Model (DREAM) (Nickovic et al., 2001; Nickovic, 2005; Nickovic et al., 2012; Vukovic et al., 2014). DREAM is an atmospheric dust mass concentration model that considers the major atmospheric processes of dust transport such as emission, horizontal and verti-
cal turbulent mixing, long-range transport and dust wet and dry deposition. The model uses the Non-hydrostatic Mesoscale Model (NMM) from the National Centers for Environmental Prediction (NCEP) as a meteorological driver. Dust particles are represented by 8 size bins with effective radii from 0.15 to 7.1 $\mu$m. For this study, the model is run with 0.1°×0.1° horizontal resolution and 28 vertical levels covering the dust-productive areas in the Sahara and the Middle East, and dust transport in the eastern Atlantic Ocean. The initial and boundary conditions for the atmospheric driver are updated every 6h from ECMWF
analysis data with 0.5°×0.5° horizontal resolution. Dust sources are defined using USGS (United States Geological Survey) land cover data combined with sediments in paleo-lake and riverine beds (Ginoux et al., 2001; Nickovic et al., 2016). DREAM model provides options for the parameterization setups of the dust-derived INPC (Nickovic et al., 2016, 2021; Ilić et al., 2021). There are several dust-dependent INPC parameterizations for immersion freezing at or above water saturation (Niemand et al., 2012; Ullrich et al., 2017; DeMott et al., 2015) and deposition nucleation at ice supersaturation (Steinke et al., 2015; Ullrich
et al., 2017). In this paper, we use the parameterizations provided by Ullrich et al. (2017) (hereafter referred to as U17) to address the immersion freezing (from -30.0 to -14.0°C) and deposition nucleation (from -67.0 to -33.0°C).

## 4 CHARACTERISATION OF ATMOSPHERIC SCENARIOS

The determination of the different atmospheric layers needed to perform the characterisation of the low and middle troposphere in Tenerife is carried out through two different and independent techniques: lidar two-layer inversions restricted to daytime
period and the corresponding atmospheric sounding (launched daily at 12 UTC). The emerging database is compiled with every lidar extinction profile and the corresponding daily sounding profile. We obtain consistent results of the MBL thickness retrieved with these two techniques as shown in Sect. S1 in the Supplementary material. This gives us confidence in the atmospheric layering performed in this study, both in terms of atmospheric aerosols and thermodynamic variables. Monthly average values of aerosol extinction at 523 nm ($\alpha$) are shown in Sect. S2 (Figs. S1 and S2 and Table S1) giving us an overview
of the dust vertical structure of the eastern Subtropical North Atlantic atmosphere throughout the year. From this analysis, we can define as predominant the clean and Saharan scenarios. Saharan scenarios present a different seasonality within the MBL, with maximum AOD values in winter, compared to the values shown in the upper layer, with maximum AOD in July and August.

ARTI (defined in Sect. 3.4) is helpful to define the clean scenarios. Clean air masses are defined as those with ARTI = 0 (i.e.,
non-African origin). Fig. S3 shows how these clean scenarios predominate at both levels through the year (see Supplementary material). Saharan air masses might significantly impact SCO from November to January (74.0 % of the annual data) while IZO is mainly affected in July and August (29.5 % of the annual data). These results corroborate our findings based on lidar results presented in Sect. S2. Based on this information, clean scenarios are split into different seasons according to the ARTI





monthly distribution, i.e., winter (Nov-Jan), extended spring (Feb-Jun), summer (Jul-Aug) and autumn (Sep-Oct). Otherwise,

Saharan scenarios in winter (Winter-SAL, confined to lower levels) are limited to the period November-January while Saharan scenarios in summer (Summer-SAL, as an elevated layer) are limited to July and August.

AERONET AOD and AE have been used to better categorise the Saharan scenarios, in addition to the previous criterion (ARTI > 0). Winter-SAL scenario is defined as those conditions in November-January with ARTI > 0, AOD at SCO $\geq 0.15$ and Angström Exponent (AE) $\leq 0.5$, according to the results from Guirado (2015). Similarly, the Summer-SAL scenario is

defined by ARTI > 0, AOD at IZO $\geq 0.10$ and AE at IZO $\leq 0.6$ (Guirado, 2015).

## 4.1  Clean scenarios

Vertical profiles of meteorological parameters ($T$, $\theta$, $\theta_e$, $r$ and horizontal wind) and $\alpha$, corresponding to the clean scenarios, are shown in Fig. 1 and Table 1. We used a total of 10658 lidar profiles corresponding to 610 days in the case of the extended spring scenario, 5341 (217 days) for summer, 3842 (240 days) for autumn, and 1532 (136 days) for winter. The MBL is displayed in

Fig. 1 as a light blue shaded layer with a horizontal blue line marking the top of the MBL, i.e., the MBI. The presence of the TWL in summer is represented as a dark blue shaded layer.

A seasonal variability of the MBL thickness is observed, extending up to 1621 m in autumn and to 1238 m in summer. However, we find a more marked seasonal dependence on the MBI's base altitude, with a minimum value in summertime (878 m) and a maximum in September-October (1326 m). This variability is strongly influenced by thermal processes at the

surface and also by large-scale subsidence processes (Carrillo et al., 2016). The intensification of subsidence processes in summertime impacts positively on the strength of the MBI (maximum strength of 3.1 K in summer). It explains the greater vertical extension of the MBL in September-October because subsidence is at a minimum and its lower vertical extension in summer when subsidence is at a maximum.

Mean and median lidar profiles (Fig. 1) show a rather constant $\alpha$ profile within the MBL, with two secondary maximum

values. The first maximum in the median profile, with $\alpha \sim 0.033$ km$^{-1}$, is located at altitudes ranging from 0.6 km in summer to 0.7 km in autumn-winter. Maximum $\alpha$ peak values are found in winter (0.036 km$^{-1}$), which may be a consequence of the residual dust due to the frequent dust outbreaks at this level in this season. The second maximum ($\alpha \sim 0.023$ km$^{-1}$) is located near the top of the MBL (at $\sim$ 1.5 -1.7 km height). Minimum $\alpha$ values within the MBL are retrieved near the MBL's top, ranging from 1.6 km in summer to 1.3 km in the other seasons. Low $\alpha$ values, below 0.022 km$^{-1}$, are observed at 2 km, in the

clean free troposphere above (minimum value of 0.016 km$^{-1}$ in winter, and maximum value of 0.022 km$^{-1}$ in autumn).



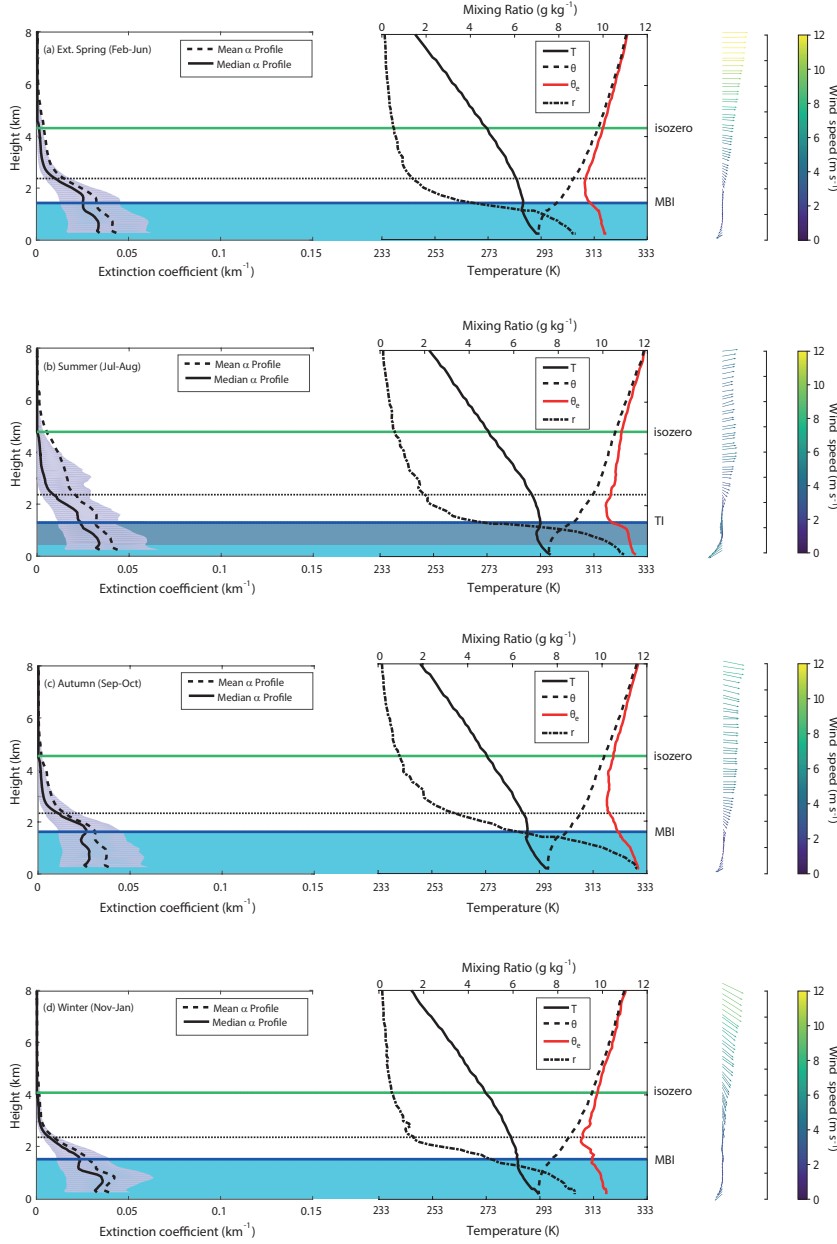

**Figure 1.** Extinction profiles (mean and median) and vertical profiles (median) of temperature ($T$), potential temperature ($\theta$), equivalent potential temperature ($\theta_e$) and water vapour mixing ratio ($r$) for (a) spring, (b) summer, (c) autumn and (d) winter clean scenarios. The black horizontal broken line represents the IZO altitude. The blue solid line represents the MBI and the TI tops detected from vertical soundings. The green horizontal line indicates the average 0°C (isozero) level extracted from vertical soundings. The grey shaded area is bounded by the $20^{th}$ and $80^{th}$ percentile values of lidar total extinction at each level. 2D-field of wind profiles with arrows (wind direction) and magnitude in m s$^{-1}$ (color bar) are also presented on the right.



**Table 1.** Main descriptive statistics of the atmospheric vertical structure extracted from atmospheric soundings (median values) for clean scenarios in each season. Standard Error of the Mean (SEM) is also included. Note that the MBI top coincides with the MBL top.

| | Extended Spring (Feb-Jun) (593 soundings) | Summer (Jul-Aug) (208 soundings) | Autumn (Sep-Oct) (231 soundings) | Winter (Nov-Jan) (132 soundings) |
|---|---|---|---|---|
| Pressure (hPa) at the base | 900 ±2 | 920±2 | 870±3 | 870±5 |
| Pressure (hPa) at the top | 870 ±2 | 880±2 | 840±3 | 850±5 |
| Height (m) at the base | 1098±21 | 878±20 | 1326±37 | 1307±51 |
| Height (m) at the top | 1360 ±22 | 1238±22 | 1621±39 | 1529±52 |
| Temperature (°C) at the base | 11.6 ±0.2 | 17.2±0.2 | 14.2±0.3 | 9.4±0.4 |
| Temperature (°C) at the top | 14.8 ±0.2 | 20.9±0.2 | 17.3±0.3 | 12.8±0.4 |
| Mixing ratio ($g\ kg^{-1}$) at the base | 7.56 ±0.08 | 10.38±0.12 | 9.63±0.17 | 6.87±0.19 |
| Mixing ratio ($g\ kg^{-1}$) at the top | 2.98 ±0.10 | 4.76±0.15 | 4.63±0.20 | 2.17±0.21 |
| Thickness (m) | 203 ±43 | 284±42 | 208±77 | 194±103 |
| Strength (°C) | 2.9 ±0.4 | 3.1±0.3 | 2.6±0.6 | 2.6±0.7 |
| Mixing ratio drop ($g\ kg^{-1}$) | -4.09 ±0.18 | -5.11±0.26 | -4.18±0.40 | -4.18±0.40 |
| 0°C level (m) | 4283±61 | 4996±37 | 4602±38 | 3868±60 |





Regarding the vertical thermodynamic profiles (Fig. 1, panels on the right), the moist MBL appears as an unstable layer (negative $\theta_e$ gradient), topped by a significant temperature inversion and a sharp change in the vertical humidity profile that defines the MBL's top. Above the top of the MBL, there is a stable layer with a slightly negative $\theta_e$ gradient, characterised by a strong reduction in the rate of vertical decrease of $r$ and a minimum value in $\theta_e$ found at its top (nearly at that altitude of IZO).

These observations show the presence of a separation between the moist air within the MBL and the dry air that lies above. This interface layer is followed by a sharp change in $\theta_e$, with a positive $\theta_e$ gradient registered from this level. A positive $\theta_e$ gradient is registered from this transition layer upwards. At higher levels, very low and steady humidity profiles with almost constant $\theta$ and $\theta_e$ indicate the thermal stratification representative of CFT conditions. The isozero level extracted from vertical soundings (green horizontal line in Fig. 1) ranges from $\sim 4.9$ km in summer to $\sim 4.1$ km in winter (see Table 1).

North-East trade winds are observed within the MBL. In the FT the wind direction rotates counter-clockwise with height. The W-NW subsiding dry airflow is the predominant wind pattern in the CFT. The presence of a second layer (the TWL) in summer above 400 m (Fig. 1 (b) shaded in dark blue) with an N-NE flow regime confirms the results of Carrillo et al. (2016). This combined MBL+TWL layer in summer can be considered as a layer of different nature than the MBL, capped by a 'transition inversion' (TI) between the lower troposphere and the FT. The maximum wind shear, calculated as the gradient of

the magnitude of the wind vector, is found between 1.2 km and 1.8 km, roughly matching the height of the MBI, and the TI in summer. These results are also consistent with the maximum vertical variation of the wind direction.

### 4.2    Saharan Scenarios

Here we perform a similar analysis for the two main scenarios of SAL intrusions that take place in summer (Fig. 2 (a)) and winter (Fig. 2 (b)). Winter-SAL scenario is included in this Saharan characterisation although it does not follow strictly the

original description of the SAL as a hot, dry, dusty layer generated over the Sahara desert. In this winter case, the dust intrusions do not constitute an elevated layer but a dust transport confined to lower levels.

We used a total of 3529 lidar vertical profiles (173 days) for summer, and 2437 lidar profiles (corresponding to 105 days) for winter. The main features of each dust scenario are shown in both Fig. 2 and Table 2.

#### 4.2.1    Saharan scenario in summer (Summer-SAL)

In this season, a deep African CBL is established over North Africa (Sahara) due to the turbulent and convective fluxes of sensible heat from the hot surface. The temperature gradient in the lower troposphere (e.g. 925 hPa) between Tenerife and a point located about 500 km above Western Sahara in summer is roughly 10 K or higher (not shown here), and a strong trade wind regime is established (Carrillo et al., 2016). This results in a cold and humid MBL that becomes an impenetrable barrier to the hot dust-laden SAL. When the SAL moves westwards over the subtropical North Atlantic, it often extends to 5 - 6

km height. Its base lifts over a cooler, denser and moister maritime air occupying the lower boundary layer, the MBL. The superposition of these two very different air masses produces substantial changes in the vertical structure of the subtropical troposphere over the North Atlantic that can be well identified by comparing Figs. 1 (b) and 2 (a).



**Table 2.** Main statistics of the atmospheric vertical structure extracted from atmospheric soundings (median values) for winter season. Standard Error of the mean (SEM) is also included.

| | Winter (Nov-Jan) (96 soundings) | | Summer (Jul-Aug) (166 soundings) | |
|---|---|---|---|---|
| | T1 | STI | T1 | STI |
| Pressure (hPa) at the base | 870 (7) | 800 (11) | 950 (2) | 500 (14) |
| Pressure (hPa) at the top | 850 (7) | 780 (11) | 920 (2) | 480 (14) |
| Height (m) at the base | 1410 (75) | 2088 (125) | 586 (22) | 5970 (197) |
| Height (m) at the top | 1572 (76) | 2314 (125) | 885 (27) | 6234 (199) |
| Temperature (°C) at the base | 9.5 (0.6) | 7.6 (0.9) | 20.6 (0.3) | -9.0 (1.4) |
| Temperature (°C) at the top | 10.7 (0.6) | 8.6 (1.0) | 25.6 (0.3) | -8.2 (1.5) |
| Mixing ratio ($g\ kg^{-1}$) at the base | 5.24 (0.29) | 2.43 (0.25) | 10.99 (0.19) | 1.20 (0.13) |
| Mixing ratio ($g\ kg^{-1}$) at the top | 2.11 (0.19) | 1.19 (0.19) | 4.49 (0.21) | 0.38 (0.13) |
| Thickness (m) | 178 (152) | 194 (250) | 283 (50) | 173 (397) |
| Strength (°C) | 1.1 (1.2) | 0.7 (1.9) | 5.0 (0.6) | 0.7 (2.9) |
| Mixing ratio drop ($g\ kg^{-1}$) | -2.39 (0.48) | -1.04 (0.44) | -5.90 (0.40) | -0.71 (0.25) |
| 0°C level (m) | 3641±80 | | 4624±55 | |

The base of the SAL reinforces the temperature inversion at the top of the MBL observed before the SAL intrusion (strength of the MBI inversion of 5 K from Table 2), causing the narrowing of the MBL (Fig. 2 (a)). In the Supplement, we show a

version of Fig. 2 that includes relative humidity ($RH$) (Fig. S4). In some cases, the MBL may be reduced to less than 500 m depth (Alastuey et al., 2005). The MBL retains its characteristics of $\theta$ and $r$, with the highest humidity confined to lower levels, with a maximum in $r$ ($\sim$ 11.6 g kg$^{-1}$) at $\sim$ 200 m. The change in wind direction at this level, from NE to N-NE, points to the presence of two different layers, the MBL and the TWL that lies above. These results are similar to those found in the clean scenario in summer (Fig. 1 (b)) and they are in concordance with the results of Carrillo et al. (2016). These two layers (MBL

and TWL) are represented in Fig. 2 as shaded in light and dark blue, respectively. $\theta$ at the surface level does not change its value of $\sim$ 297 K found in summer clean conditions, although $\alpha$ increases slightly up to values of 0.049 km$^{-1}$, a 48 % higher in reference to the $\alpha$ maximum found in the summer-clean scenario at surface level (0.033 km$^{-1}$). This residual dust found in the MBL underneath the SAL, also reported by other authors (i.e. Reid et al., 2003), is a result of turbulent mixing caused by wind shear in the transition region between the MBL and SAL (Colarco et al., 2003a, b; Rittmeister et al., 2017). The increase

in temperature and the drying of the MLB top causes that the condensation level is no longer reached at the MBI height, dissipating the sea of clouds (stratocumulus) (see Sect. 7). A pronounced decrease in atmospheric moisture (lower $r$ values by 3.5 g kg$^{-1}$) and a negative $\theta_e$ vertical gradient is found above the MBI, with a minimum $\theta_e$ value at $\sim$ 1.1 km, matching the





MBL's top. This level coincides with the minimum aerosol concentration and, as a result, indicates the limit between different air masses.

Above the MBL, the SAL maintains unchanged much of its original characteristics acquired when it was formed over the Sahara, even though its base has risen above the MBL and its top is capped by a temperature inversion (STI) (Carlson, 2016). The STI is located between 6.0 and 6.2 km in altitude. This feature is also confirmed by the $\theta$ vertical gradient, indicating 6.0 - 6.5 km height as the end of the well-mixed SAL layer, coinciding with the height of neutral or slightly positive $\theta$ and $\theta_e$ vertical gradients. The warm SAL maintains an almost constant $\theta_e$ at $\sim$ 327 K and $\theta$ at $\sim$ 318 K over a large part of its vertical

extension, from 2.5 to 5 km height, which is the typical temperature of the mixing layer over the Sahara. The $\theta$ value within the SAL is consistent with those found by other authors, $\sim$ 316 K by Carlson (2016), $\sim$ 317 K by Carlson and Prospero (1972); Garcia-Carreras et al. (2015), and $\sim$ 315 K by Gutleben et al. (2019). In terms of aerosol load, the SAL likely maintains a similar dust content than that when it was formed over the continent, with aerosol extinction coefficients $>$ 0.065 km$^{-1}$. The maximum aerosol load is located slightly higher than the IZO level (2.5 km, 0.066 km$^{-1}$). A sharp decrease in $\alpha$ is observed

near the top of this layer in coincidence with an increase in the $RH$ profile (Fig. S4), with a maximum of 47 % at 5.2 - 5.5 km height, $\sim$ 1 km below the STI level. It is noteworthy that the SAL has a water vapour content much higher ($\sim$ 2 g kg$^{-1}$) than that of the clean free troposphere over the subtropical North Atlantic, as Andrey et al. (2014) had reported.

So, in terms of its vertical features, the SAL is characterised by a well-mixed layer with a generally fairly constant potential temperature, vapour mixing ratio, dust particles concentration and size distribution with height (Maring et al., 2003). This

relatively unchanged vertical thermodynamic structure, a consequence of the persistent temperature inversions located at the top and the base of the SAL, colder (near the top) and warmer (at the base) than its surroundings (Jury and Whitehall, 2010), and the vertical mixing processes within the layer (Gasteiger et al., 2017), explains the longevity of this air mass and the higher- than-expected retention of coarse-mode particles during the long-range transport. According to Ryder et al. (2018) and references therein, this internal mixing is hypothesised to be sustained by vertical convection (mixing) by solar absorption, turbulence,

or electrostatic charging. This, together with the fact that the subtropical North Atlantic troposphere is basically barotropic in summer, means that the SAL can travel thousands of kilometres while maintaining unaltered its main thermodynamic properties and aerosol content.

Regarding the isozero level, our results indicate that this isotherm altitude is found 190 m lower than in the summer-clean scenario, indicating the presence of colder air at higher levels compared to the clean scenario.

The wind regime within the MBL corresponds to marine-aerosol-laden trade wind flow (NE), while the dust-laden SW airflow is predominant within the SAL layer. Similarly to the clean scenario in summer (Fig. 1 (b)), two different wind regimes are found in the lower levels, pointing to the presence of both MBL and TWL below the SAL. Maximum wind values are found at altitudes slightly below 6 km, coinciding with the maximum humidity change within the SAL layer (reduction of 63 % in $r$ values between 5.5 km and 6.3 km, the STI height). The Summer-SAL is also identified by a counter-clockwise wind rotation

(warm advection) of winds near the $\alpha$ peak. In fact, maximum shear in wind direction is found at 2.3 km, and at 1.6, 2.5 and 6.3 km in the case of the maximum shear in the wind speed, matching well the height of the lower part of the SAL, the dust peak, and the STI, respectively.

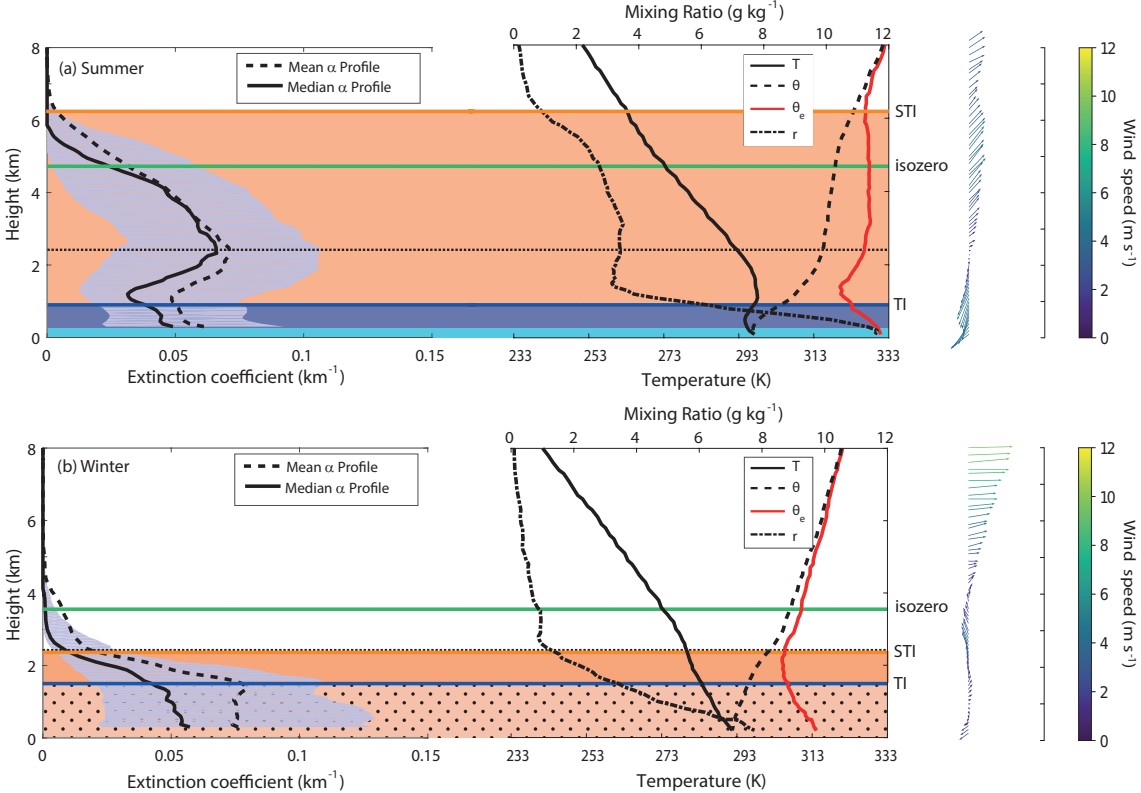

**Figure 2.** The same as Fig. 1, but for Saharan scenarios in (a) summer and (b) winter. Dark and light blue areas in summer indicate the TWL and the MBL, respectively. Orange shaded areas indicate the presence of the SAL. Orange shaded areas with black spots in winter indicate a mixture of the SAL with the MBL.

Some of the characteristics of the lower-middle troposphere vertical profile over the subtropical North Atlantic region in summer will be better understood once the SAL's impact is analysed in Sects. 5 and 6.

### 4.2.2   Saharan scenario in winter (Winter-SAL)

The winter dust intrusions over the subtropical eastern North Atlantic region are characterised by the presence of a relatively thin SAL reaching lower altitudes than the Summer-SAL (up to about 2 km height). In this season, the temperature difference in the lower troposphere (e.g. 925 hPa) between Tenerife and a point located about 500 km above Western Sahara is barely 2 K (not shown here), and the flow associated with the trade wind regime is very weak (Carrillo et al., 2016). So that the Winter-SAL penetrates the MBL mixing dust with marine aerosols.

Two different layers are observed in the vertical extinction profile in Fig. 2 (b) below the IZO level: the MBL mixed with dust below the MBI and a relatively narrow layer of the original SAL just above this transition layer. The first $\alpha$ maximum, located at 1.2 km, presents median extinction coefficients of 0.057 km$^{-1}$ that corresponds to a mixture of mineral dust with





marine aerosol, considerably higher than the maximum value found at these levels in the clean scenario in winter ($0.036 \, \text{km}^{-1}$

as seen in Fig. 1 (d)). It can be seen that dust conditions were also affecting SCO, with the extinction coefficient being almost two times higher than the value found in clean conditions ($\alpha$ of $0.032 \, \text{km}^{-1}$ as seen in Fig. 1 (d)).

From the thermodynamic profiles, we observe that the MBL contains the maximum mixing ratio ($r \sim 7.76 \, \text{g kg}^{-1}$) located at surface level. It suggests the atmosphere at lower levels is considerably drier than in clean conditions in winter when we found a peak in $r$ of $8.76 \, \text{g kg}^{-1}$ at similar levels (reduction of 11 % relative to clean conditions). This layer presents a nearly

stable $\alpha$ ($0.057 - 0.049 \, \text{km}^{-1}$) and is followed by a transition layer capped by a strong change in the vertical humidity gradient. The second elevated layer is associated with the impact of the SAL on the lower FT with maximum $\alpha$ values ($\sim 0.049 \, \text{km}^{-1}$) located at 1.3 km height. A continuous decrease in atmospheric moisture with altitude ($r$ values of $1.57 \, \text{g kg}^{-1}$ in the MBI) is observed in this layer, with a minimum in $\theta_e$ observed at 2.3 km.

The weakness of the temperature inversion in this scenario (only 1.1 K) denotes there is no marked decoupling between

layers. We note the significant difference between mean and median $\alpha$ values at all levels, pointing to the occurrence of SAL intrusions of very different intensities. We have not found a significant reduction in the isozero level in comparison to the clean scenario in winter.

Wind vectors show the presence of NE wind component at the surface (Fig. 2 (b)), indicative of the trade wind regime, followed by a cold advection (anti-clockwise rotation of wind with height) characterised by SW airflows laden with Saharan

mineral dust within the SAL layer. High values of the wind shear are found at 2.4 km, coinciding with the STI. SE winds are observed to be the dominant pattern in CFT conditions up to 4 km, and the normal W pattern is retrieved above this level.

## 5 SAL's impact on the aerosol extinction and the thermodynamic profiles in the summer/winter troposphere

In order to understand the impact of the SAL on the subtropical atmospheric vertical structure we have displayed in Fig. 3 the differences in $T$, $r$ and $\alpha$ profiles between the summer/winter Saharan scenarios and their corresponding clean scenarios. The

magnitude of the previous differences depends on the seasonal values of these parameters in the subtropical troposphere in addition to their seasonal values of the advected Saharan air mass.

In the case of the Summer-SAL, these differences are readily appreciated, up to 6 K warmer at lower levels (Fig. 3 (a)), in response to both the SAL radiative impact (Sect. 6) and the warm advection of dusty Saharan air masses. Negative temperature differences of up to 2 K are found at 5.2 km. This cooling effect results in an inversion at the Summer-SAL top corresponding

to the upper limit of the dust layer. In contrast, we can describe the Winter-SAL, when dust transportation is confined to lower levels, as a layer with negligible impact in terms of $T$ at lower levels (Fig. 3 (d)). However, the $T$ profile from 2 km up is roughly 3 K cooler than in clean conditions.

Our results show that the SAL has also a strong impact on atmospheric humidity (Figs. 5 (b) and (e)), similarly to the results presented by Dunion and Marron (2008); Andrey et al. (2014). Our results present the SAL as a dry layer at lower levels in both

summer and winter. We observe negative $r$ anomalies (Fig. 3 (b)) at the SAL's base in the case of the Summer-SAL ($\Delta r \sim$ -3.6 g kg$^{-1}$) and slightly higher $\alpha$ anomalies ($\Delta \alpha \sim +48$ %) compared to clean summertime conditions. Regarding higher levels,





positive and rather constant $r$ anomalies of about 2 g kg$^{-1}$ are found in the Summer-SAL from the dust peak up to 5 km. These $r$ anomalies, although they may seem small, represent a high relative variation in comparison to the dry free troposphere under clean conditions ($\Delta r$ up to +332 % at 5.3 km, roughly coinciding with the altitude of the maximum cooling). The anomaly

in $\alpha$ peaks at 2.6 km, where maximum extinction values are commonly retrieved (Fig. 3 (c)). As a consequence, the SAL appears as a moist layer at mid-levels because this layer transports relatively moist air at altitudes where atmospheric humidity is commonly very low. The Winter-SAL at lower levels is characterised by mixing ratio anomalies up to $\sim$ -3.0 g kg$^{-1}$ (dry anomaly) below the SAL ($\Delta r$ of -38 % at 1 km) (Fig. 3 (e)), similar to those observed in the Summer-SAL at similar levels. High $\alpha$ anomalies were also observed in the MBL (maximum values of $\Delta \alpha \sim$ +118 % located in the dust peak level) (Fig. 3

(f)), with any significant anomaly in terms of mixing ratio nor $\alpha$ at higher levels.

The fact that the SAL in summer presents a significant and positive anomaly in water vapour content in the subtropics, as shown in this study, is consistent with previous results (i.e. Andrey et al., 2014; Kim et al., 2004; Gutleben et al., 2019, 2020, among others). This result shows that, despite being a dry layer, the SAL in the subtropical region is more humid than the clean FT around it. This fact is relevant since water vapour is a gas strongly absorbing long-wave radiation. Consequently, moisture

carried on the SAL can play a key role in the atmospheric radiative balance, depending on its vertical distribution, and therefore in the temporal evolution of the vertical structure of the SAL itself, as discussed in Sect. 6.

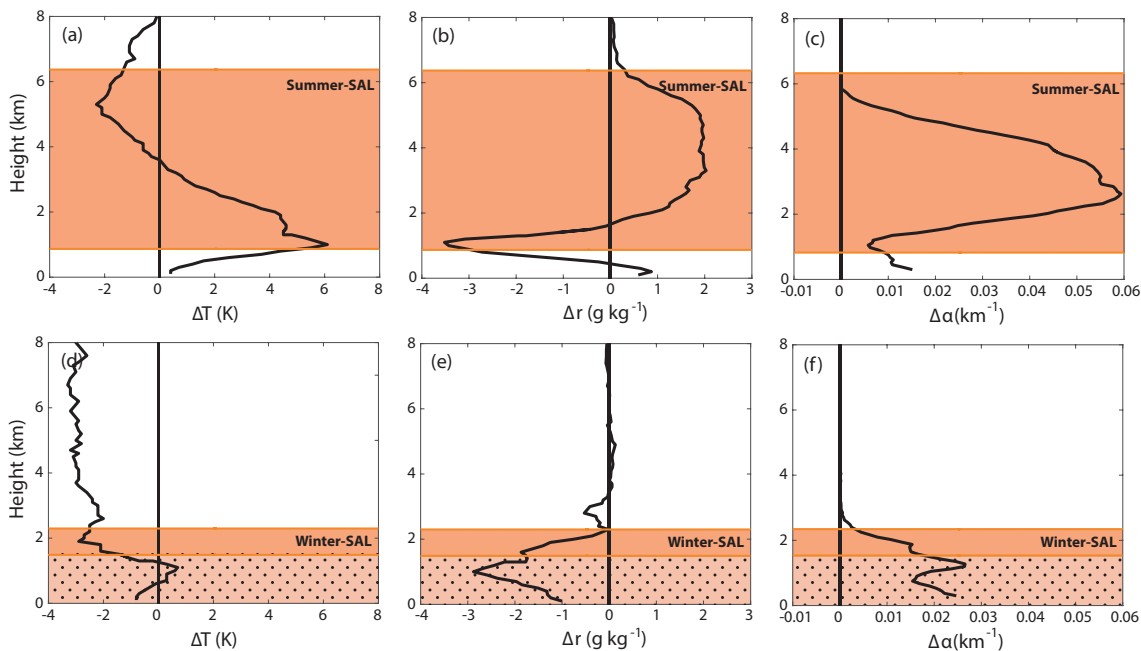

**Figure 3.** Differences between Saharan and clean scenarios in summer and winter in terms of temperature ($T$) and water vapour mixing ratio ($r$), both magnitudes retrieved from radiosondes; and extinction coefficient ($\alpha$) from MPL elastic backscatter lidar. Orange shaded areas indicate the presence of the SAL. Orange shaded area with black spots in winter, indicates a mixture of SAL with MBL.



# 6 SAL's impact on the vertical atmospheric heating rates

Several papers in the literature have been focused on determining the SAL radiative effects in summer (Carlson and Benjamin, 1980; Kim et al., 2004; Wong et al., 2009; Chen et al., 2010; Gutleben et al., 2019, 2020). These authors found that, under clear sky conditions, the net radiative effect, a combination of LW and SW interactions, results in maximum net heating located slightly below the maximum dust concentration level, and also a minor net heating observed near the surface. The existence of significant heating rates ($\sim 1$ K day$^{-1}$ between 500 hPa and 1000 hPa) indicates the important role of dust in stabilising the atmosphere, as, for example, the enhancement of a pre-existing trade wind temperature inversion found by Chen et al. (2010) as a result of the SAL layer above. In addition, cold anomalies above the SAL are hypothesised as the result of the strong vertical ascend of warm air due to the net warming within the SAL, yielding to an adiabatic cooling in these upper layers (Dunion and Marron, 2008; Wong et al., 2009; Chen et al., 2010). However, other authors (Kim et al., 2004; Gutleben et al., 2019, 2020) attributed an important role in the aforementioned vertical radiative heating rates to the enhanced atmospheric water vapour within the SAL. As a result, both dust and dry anomalies play an important role in heating of the SAL in the lower troposphere (Wong et al., 2009).

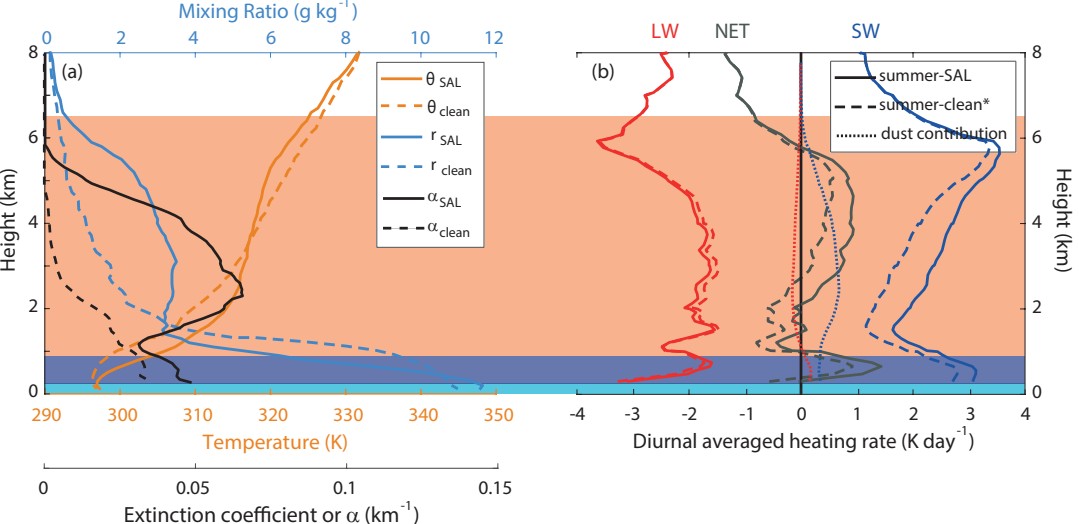

**Figure 4.** (a) Median potential temperature ($\theta$) and water vapour mixing ratio ($r$) profiles extracted from the radiosondes, and median lidar extinction ($\alpha$) profiles, for summer-clean and Summer-SAL scenarios. (b) Diurnal averaged heating rates simulated with Libradtran using MOPSMAP inputs for the scenarios Summer-SAL (solid lines) and summer-clean* (dashed lines). Dust contribution to total heating is also included with dotted lines. The orange shaded area indicates the presence of the SAL, while the dark and light blue areas represent the TWL and the MBL, respectively.

Atmospheric heating rates have been calculated using the radiative transfer model LibRadtran (see Sect. 3.5) in order to detect the vertical radiative impact of the dust layer in summer and winter. As we have shown in the previous section, this dust-laden layer acts significantly modifying the thermodynamic profiles in the lower-middle troposphere.


Diurnal-averaged heating rates have been retrieved for two different cases. The first case, representative of the real atmospheric conditions under the presence of the Summer- or Winter-SAL, includes the median $T$, $r$ (or $RH$) and $\alpha$ representative of each scenario (Sect. 4.2.1 and 4.2.2, respectively). A second case has been considered including the same median thermodynamic soundings as in the previous dusty scenarios but using the median $\alpha$ profiles representative of clean conditions (retrieved in Sect. 4.1). These scenarios, called summer-clean* and winter-clean*, are not real but can be used to separate contributions of mineral dust and water vapour to the radiative forcing in the atmospheric column. This approach is similar to that carried out in previous studies (Kim et al., 2004; Gutleben et al., 2019, 2020).

Simulations have been done including in LibRadtran spectral information of $\alpha$, $SSA$ and $g$ extracted from MOPSMAP (Gasteiger and Wiegner, 2018). In the case of both Summer-SAL and Winter-SAL scenarios, a mixture of maritime clean and desert dust aerosol types have been considered to form the MBL, and only desert dust aerosols are considered to form the atmosphere above this layer. In the case of the summer-clean* and winter-clean* scenarios, only maritime clean aerosols have been considered. MOPSMAP spectral resolution of these optical aerosol properties depends on the spectral range, being modelled between 400 and 1000 nm with a spectral resolution of 10 nm, between 1000 and 3000 nm with a resolution of 500 nm and finally, with a spectral resolution of 3000 nm in the range 3000 nm to 40 $\mu$m. Spectral AE retrieved from MOPSMAP has been used to calculate the $\alpha$ profiles at the other wavelengths by applying the Angstrom power law (Ångström, 1929) once the vertical profile from the lidar at 523 nm is known.

Fig. 4 (a) shows $T$, $r$ and $\alpha$ profiles for the summer clean and the Summer-SAL scenarios, while Fig. 4 (b) presents the comparison of the modelled output heating rates (in K day$^{-1}$) for the same scenarios. The maximum heating rate due to mineral dust (differences between the Summer-SAL and the summer-clean* atmospheres) is +0.7 K day$^{-1}$ and -0.15 K day$^{-1}$ for SW and LW (dotted lines in Fig. 4 (b)), respectively, located at about 2.5 and 3 km, near the dust concentration peak. Following Chen et al. (2010), the net radiative effect near the surface is a balance between absorption heating, extinction cooling, emission cooling, and adiabatic heating/cooling. In our case, we have found net heating near the surface dominated by the SW forcing, attributed to the net effect of mineral dust (SW absorption and LW cooling). This result is consistent with previous studies in the literature (Carlson and Benjamin, 1980; Kim et al., 2004; Wong et al., 2009; Chen et al., 2010; Zhu et al., 2007) who found an expected and dominating dust-induced SW radiative warming slightly below the dust maximum. Below this dust peak, a weaker SW effect is expected, dominated by dust extinction processes. We have actually observed that LW dominates below the dust peak through upwards and downwards emission processes (Zhu et al., 2007). The mean net heating rate due to mineral dust observed in the first 1 km height is +0.34 K day$^{-1}$ and +0.13 K day$^{-1}$ for SW and LW, respectively. The LW warming found at surface level has been also observed by Chen et al. (2010), who attributed it to the combined effect of dust LW absorption, reflection and re-emission processes at lower levels. Above 6 km the heating rate due to aerosols does not show any change due to the absence of dust. However, in view of Fig. 4 (b), one can see how the real Summer-SAL radiative impact is characterised by a strong cold anomaly near the SAL's top which cannot be attributed to the solely radiative effect of mineral dust. This cold anomaly, also observed by other authors (Dunion and Marron, 2008; Wong et al., 2009; Chen et al., 2010; Gutleben et al., 2019, 2020), are consistent with the temperature difference of about -2 K (Fig. 3 (a)), and also by the 190 m descent from isozero height level (Tables 1 and 2). Fig. 4 (b) quantifies the net Summer-SAL





heating rate to be -0.6 K in the SAL's top. Maximum LW cooling (-3.6 K day$^{-1}$) is observed at 5.9 km while maximum SW heating (3.5 K day$^{-1}$) is observed slightly below, at 5.6 km, both located near the maximum $RH$ peak within the SAL (see Fig.

S4). Net warming is observed to occur within the SAL (from 2.1 km to 5.7 km), with relatively constant heatings rates of about 0.8 K day$^{-1}$ in an altitude range between 3.5 km and 5.1 km. Atmospheric water vapour absorbs part of the SW radiation, with a maximum effect found in the position of the $RH$ peak within the SAL (5.6 km). LW cooling due to water vapour is attributed to the LW emission without absorption (adiabatic cooling) in the top of the SAL (maximum cooling found at 5.9 km), induced by the strong vertical ascent of air near the STI, acting as a thermodynamic layer of discontinuity between two different air

masses.

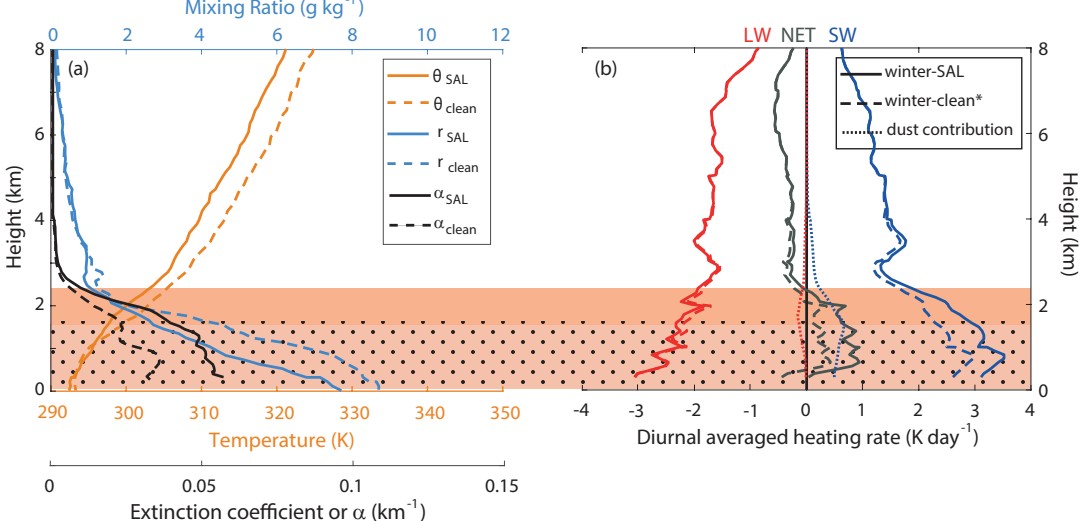

**Figure 5.** The same as Fig. 4, but for winter-clean and Winter-SAL scenarios. The orange shaded area indicates the presence of the SAL, while the orange shaded area with black spots indicate a mixture of the SAL with the MBL.

     Our results highlight the important radiative role played by the water vapour on the atmospheric column in the Summer-SAL scenario, not only in the cold anomaly near the STI but also at lower levels counteracting the warming effect of aerosols in the SW. This is a relevant aspect to consider, especially in atmospheric modelling. Dust models might certainly introduce an overestimation of the aerosol radiative effect within the SAL if the water vapour radiative effect is obviated.

The same analysis in terms of modelled output heating rates (in K day$^{-1}$) has been performed for the winter clean and winter-dust scenarios (Fig. 5). As in the previous case of summer, the radiative effect of dust has been estimated by means of the heating rates differences between Winter-SAL and winter-clean* atmospheres. We have observed a dust-induced radiative effect dominated by SW heating (maximum heating of +0.7 K day$^{-1}$ at 1.5 km, near the dust peak), while below the dust peak the SW effect becomes lower. LW cooling is also maximum near the $\alpha$ peak (-0.13 K day$^{-1}$). However, the real Winter-SAL

radiative impact presents maximum SW heating rates up to +3.5 K day$^{-1}$ at 0.8 km, near the MBI, while the maximum LW cooling is found near the surface (-3.1 K day$^{-1}$) (in Fig. 5). The net effect within the surface layer (MBL and SAL) implies





a small and rather constant SW warming of about +0.7 K day$^{-1}$ in the 0.6-2 km layer. This net heating near the surface is attributed to the combined effect of dominant dust SW heating and the predominant effect of water vapour SW absorption at these levels. Above the SAL, the net radiative effect is dominated by the dry anomaly of the Winter-SAL in comparison to
clean conditions. This removal of water vapour in the first 2 km reduces the greenhouse trapping of LW radiation, with an LW cooling effect of about +0.3 K day$^{-1}$ (Wong et al., 2009). As a result, dry anomalies in the first levels have the consequence of thermal cooling (decrease in the net upward longwave flux) from the top of the SAL upwards. In this winter case, the SAL also impacts the atmospheric column, from the dry anomaly at lower levels to the thermal cooling at higher levels.

        The previous findings in addition to those on Sect. 5 allow us to assess the role of the thermal advection on the differences
observed on the vertical temperature profiles taking place within the Summer- and Winter-SAL. According to Fig. 3 (a), a temperature difference between the Saharan and the clean scenarios greater than 3 K occurs between 0.8 m and 2.5 km height, registering the maximum temperature increase in the presence of SAL at $\sim$ 1 km height, with a $\Delta T = +6$ K. This height is precisely where the maximum net radiative forcing cooling is recorded in the lower troposphere (-0.5 K day$^{-1}$ as shown in Fig. 4). This result might lead us to conclude that the warm air advection from the Sahara is the major driver of the
temperature increase observed in this altitude interval. An analysis based on composites of average potential temperature from the NCEP/NCAR Reanalysis (Kalnay et al., 1996) in summer (July and August) in the time period 2007-2018 has served us to estimate the thermal advection at the Canary Islands from the warmest region of the Sahara, located over west-central Algeria around the Greenwich meridian. This analysis (shown in Supplement Fig. S5 (a)) estimated an increase in temperature of up to + 9 K at 1 km height due to this warm advection. Thus, we can say that the advection dominates between 800 m and 2.5 km
height, but the radiative processes are more important below 800 m in the MBL (heating), between 2.5 and 6 km (heating), and above 6 km (cooling). In winter, there is practically no temperature gradient between the ocean and the African continent in the troposphere (Fig. S5 (b)), so temperature advection processes do not take place. A more realistic separation of the effects of temperature advection from that of radiative processes would be possible with dust model simulations that allow for the activation and deactivation of the radiative processes of mineral dust and water vapour, a capability that is out of the scope of
this paper.

## 7    Possible SAL's impact on cloud formation in the subtropical troposphere

As we have shown in Sects. 5 and 6, both SAL's dust and humidity anomalies profoundly change the vertical radiative fluxes. The statistics of clouds detected in each dust scenario have been extracted from MPL-3 data in order to investigate the effect of SAL radiative impact on cloud formation. Because the MPL system used in this study cannot measure particle depolarization,
we are not able to directly estimate the presence of clouds and their respective altitudes. We addressed this issue by means of an indirect technique. We first extracted extinction coefficient values assuming a lidar ratio of 20 sr, close to the expected value for mid-level clouds (Yorks et al., 2011). Then, we have identified molecular signal (particle-free and therefore with the sole contribution of molecules) and aerosol signal taking into account the percentile, the size and also the value of the retrieved $\alpha$. The presence of cloud is considered when visibility is reduced to values < 5 km if the molecular layer starts from the surface



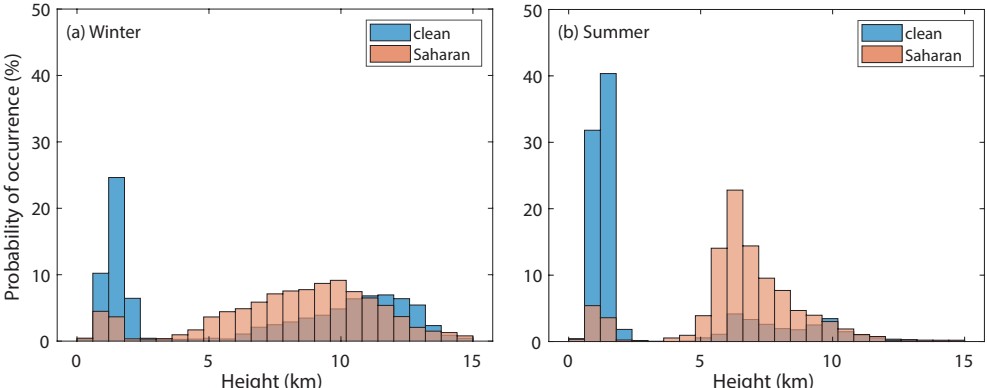

**Figure 6.** Probability of occurrence of clouds detected by the MPL-3 lidar in the vertical column in (a) winter and (b) summer seasons under clean and Saharan conditions.

level, according to the World Meteorological Organization criteria (WMO, 2019), or when visibility is reduced to values $< 10$ km if the molecular layer starts above the surface level. We present the probability of occurrence of clouds extracted with this classification analysis for both the Saharan and the clean scenarios, for winter (Fig. 6 (a)) and summer (Fig. 6 (b)). According to WMO, low clouds are usually located between surface and 2 km, middle clouds between this level and 7 km (8 km) in midlatitudes (tropics) and high clouds are usually located above this level.

A notable reduction of low clouds in the case of the two Saharan scenarios (winter and summer) in comparison to clean conditions, and simultaneously a higher occurrence of mid-level clouds under the Summer-SAL scenario are observed (Fig. 6). Although the reduction in low-level clouds is observed for both Winter-SAL and Summer-SAL, the reduction is more prominent in summer, when the probability of occurrence of low clouds is reduced from 40 % in summer clean conditions to 5 % under SAL conditions. In winter, this drop in low clouds is less pronounced, with a reduction from 25 % (winter

clean conditions) to $\sim 4$ % (Winter-SAL conditions). This observed reduction is explained taking into account that the SAL transports dryer air at lower levels (both in summer and winter), and therefore both the lifting condensation level and the level of free convection rise, increasing the energetic barrier to convection (Wong and Dessler, 2005). This low-level warming affects both summer (1.4 K day$^{-1}$) and winter (0.8 K day$^{-1}$) radiative fluxes, inhibiting the occurrence of convection, causing vertical changes or dissipation of clouds. However, there are some differences between cloud suppression in the summer and

winter SALs. In summer, SAL not only carries drier air but also warmer air masses, compressing the MBL in the first kilometre because of the enhanced subsidence processes at this time of the year. In winter the occurrence of low-level clouds is lower than in summertime due to a weaker TWL (Carrillo et al., 2016; Azorin-Molina et al., 2018). Consequently, the SAL effect is expected to be lower in winter. The result is a Saharan air mass with a strong suppressive influence on cumulus convection above the moist trade-wind layer, with a more prominent effect in summer (Carlson and Prospero, 1972), which can cause dust

to be transported into the MBL due to entrainment at the base of the SAL much more rapidly than the transport caused by the gravitational settling.



We have also observed in Fig. 6 (b) the higher occurrence of mid-level clouds under the Summer-SAL influence (23 % under Saharan conditions versus 4 % in clean conditions). This efficacy of dust to act as ice nucleation particles (INP) increases with particle size (Weinzierl et al., 2017). We can interpret some of the previous results in terms of the impact of dust on heteroge-
neous nucleation. We know that nucleation of ice particles in the atmosphere occurs by homogeneous freezing conditions (at T ≤ -37°C) in the absence of particles to catalyze ice formation, and also by heterogeneous nucleation processes due to the presence of ice-nucleating particles. This last process can occur at any temperature below 0°C and involves typically mid-level clouds composed of ice crystals and supercooled water droplets (Murray et al., 2012). Saharan dust is known to be capable of glaciating clouds at temperatures unusually warm, between -5°C and -9°C (Sassen et al., 2003), being this characteristic layer,
consisting typically of altocumulus and altostratus, commonly found at the top of the near-adiabatic part of the profile, i.e., in the upper region of the SAL (Sassen et al., 2003; Parker et al., 2005). In our work, clouds observed under the Summer-SAL influence are predominantly mid-level clouds (Fig. 6 (a)), mostly located between 6 km and 7 km height, coinciding with SAL's top (STI). Conversely, we have not observed any significant feature in the case of the comparison between clean and dusty scenarios in winter at mid-levels. These mid-level clouds under Summer-SAL conditions are detected after the peak in
$RH$ (45 %-50 %) associated with the SAL layer (see Fig. S4) and also at altitudes higher than the isozero level (between 5 km and 7 km). Temperatures at these altitudes are relatively warm, approximately between -9°C and -15°C according to the median temperature profile in Fig. 2 (a), which is in agreement with previous studies (Sassen et al., 2003).

As a result, we have found an indication of a possible effect of dust on heterogeneous ice nucleation, resulting in modestly supercooled mid-level clouds at altitudes where moisture and temperature conditions are favourable, i.e., near the STI. This
enhancement of mid-level clouds when dust is present may have a significant impact on the atmospheric radiative balance.

To further investigate the role of dust in cloud formation, a case study of a dust event that impacted Tenerife in summer is simulated by the dust regional DREAM atmospheric model (see Sect. 3.6). The selected event on 19-24 August 2015 is associated with mid- and high-level clouds (> 5km) as it is observed in the MPL-3 profiles (see Fig. 7 (a)). MPL-3 displays a dust plume constantly influencing the station from the second half of August 19 until the first half of 22 August. Dust is
definitively detected up to 6 km, with the presence of clouds observed during the entire event from 5 to 11 km (also observed from the Cloud–Aerosol Lidar with Orthogonal Polarization -CALIOP- total attenuatted backscatter (Winker et al., 2009) in Fig. S6). INPC from the DREAM model (see Fig. 7) is calculated based on the U17 parametrization described in Sect. 3.6 (Ilić et al., 2021). The model predicts that most of the dust mass is in a layer between 2 and 6 km, in agreement with the MPL-3. Maximum concentrations (>100 $\mu g/m^3$) were reached on the afternoon of 20 August (see Fig. 7 (b)). The lidar quick-
look indicates subsidence of the dust layer and a boundary layer intrusion. Lowering of the height of peak concentration was predicted by the model, as well. However, at the top of the layer, at heights above 6 km, where temperatures are below -10°C, dust concentrations of around 5 $\mu g/m3$ can serve as a reservoir for INPs. The INPC model captures some of the features of the clouds forming above the station. Due to the temporal resolution of model outputs (3h), some features of cloud development are not seen. The predicted INPC values range from 1 to 10 $l^{-1}$, reaching values of 100 $l^{-1}$, in agreement with results of
previous remote sensing and modelling studies (Ansmann et al., 2019; Marinou et al., 2019; Ilić et al., 2021). The clear overestimation of dust concentrations by DREAM (Figs. S7 and S8 in the Supplementary material) suggests an overestimation



**Figure 7.** (a) Range-corrected MPL-3 signal (in arbitrary units) of a dust plume over Tenerife in August 2015; (b) DREAM predicted $\log_{10}$ (nINP) (in $l^{-1}$) during the same dust episode over Tenerife. Green contour represents 5 $\mu g\,m^{-3}$ dust concentration. Dashed red lines represent isotherms (in °C).



in INPC as well. Ilić et al. (2021) found that overestimation by 100 % in dust concentration results in an order of magnitude overestimation in INPC prediction. Low humidity and temperature limit at which deposition nucleation parameterization is used, indicate that this otherwise significant dust concentration does not contribute to INPC. Relatively warm mid-clouds (T>-5°C) observed by the lidar between 4 and 6 km (Fig. 7 (a)) on 20 August were not predicted by DREAM because parameterizations are done for temperatures lower than -5°C. The observed mid-level clouds were well captured by the model and were explained by indications of a local maximum in the INPC. The cirrus clouds present on 20 and 22 August are consistent with INPC values of $1\text{-}10\,l^{-1}$ and above $10\,l^{-1}$. DREAM also predicts INPC on August 22 (from noon time) and 23 but the low cumulus clouds prevent observations of mid-level and cirrus clouds. However, observations of clouds at Izaña with an all-sky camera proved the presence of cirrus from 0 am to 5 am (not shown here). These high clouds predicted by DREAM at the beginning of August 23 are also slightly discernible from the lidar observations between the low clouds (see Fig. 7 (a)). The deposition nucleation parameterization requires supersaturation with respect to ice, a condition that is not reached above 9 km. Therefore, the predicted INPC is mostly due to immersion freezing mode according to the model. The convergence between lidar observations and dust model predictions allow us to link, at least in this one instance, Summer-SAL conditions with the formation of mid- and high-level clouds. However, an extended observation period will be required to firmly establish this relationship.

## 8   Conclusions

There is a vast literature that describes the evolution of the SAL as dust outbreaks emerge from Africa and move into the North Atlantic. However, these studies lack a focus on the vertical characterisation of the Saharan layer using robust long-term datasets, a factor that limits our understanding of the atmospheric processes involved and it hinders the validation of atmospheric numerical models. Also, little attention has been paid to the seasonal variations and the radiative impact that the SAL exerts on the vertical atmospheric profiles.

Here we use long-term cloud-free lidar and vertical sounding observations performed in Tenerife during 2007-2018 to characterise the vertical structure of the lower and middle troposphere over the Eastern subtropical North Atlantic. Tenerife is a key location in Saharan dust studies because of its proximity to the most important mineral dust sources. Also, Tenerife experiences the full seasonality of dust transport at this latitude, ranging from the clean normal flow to extreme dust-loaded Saharan air mass outbreaks that affect the lower atmospheric levels in winter and higher levels in summer. The study of the SAL in the North hemisphere subtropical region will add important additional information since this location is representative of the almost pure Saharan dust. The results of this study have revealed that, in the case of clean conditions, the MBL is relatively well-mixed with $\alpha$ values $\sim 0.030$ km$^{-1}$, and significantly clean conditions are observed in the free troposphere ($\alpha < 0.022$ km$^{-1}$). We have observed a readily visible inversion layer, the MBI, with a secondary $\alpha$ maximum, which separates moist and marine aerosol-laden MBL from the clean FT above.

A clear influence of dust is observed in the Summer-SAL scenario. An important increase in $\alpha$ and a decrease in $r$ was found in the MBL, with $\alpha$ values 48 % higher and $r$ values 44 % lower relative to clean conditions. The subtropical FT was observed



to be significantly affected by the CBL. The SAL layer appears as a well-stratified layer according to the neutral or slightly positive $\theta$ and $\theta_e$ vertical gradients, with a peak in $\alpha > 0.066$ km$^{-1}$ at $\sim 2.5$ km. Under SAL influence, the CBL reaches altitudes normally characterised by clean free troposphere conditions and Saharan air masses are more humid than FT at the same levels with a peak in $r$ near the SAL's top ($\Delta r \sim +332$ % at 5.3 km). It is noteworthy the decrease in the isozero level in this Saharan scenario, in comparison to the clean scenario in summer, indicating the presence of colder air at higher levels

under the presence of SAL. Its top, the STI, has been located at $\sim 6.0$ km. It is important to highlight the reinforcement in the MBI as a result of the SAL effect, with a strength of 3.1 K in the clean scenario in summer and 5.0 K in the same season with dust influence. In the winter-Saharan scenario, in which dust is only present at lower levels, we have found two different layers in the MBL but compressed in the first 2 km height: a dry ($\Delta r \sim -38$ % in comparison to the clean scenario) and well mixed MBL and the SAL with extinction coefficients up to 0.049 km$^{-1}$ ($\Delta \alpha$ maximum of +118 %) in the SAL's peak ($\sim 1.3$ km

height). CFT conditions were found from 2.3 km on.

Important anomalies in comparison to clean conditions were observed in the lower-middle troposphere as a consequence of the warm and moist advection attributed to the Summer-SAL. This layer appears as a warmer layer ($\Delta T$ up to +6 K within the SAL) and colder at its top ($\Delta T$ up to -2 K near the SAL's top); drier at the base ($\Delta r \sim -44$ %) but more humid within the SAL ($\Delta r \sim +332$ % at 5.3 km). In the Winter-SAL scenario, a net cooling of $\sim 3$ K is observed above 2 km height. In this case, the

SAL appeared as a drier ($\Delta r \sim -38$ %) and dustier ($\Delta \alpha$ maximum of +118 %) layer, in comparison to winter clean conditions.

We have delved into the previous thermal and moisture anomalies by means of radiative transfer calculations using a 12-year climatology of lidar and thermodynamic profiles. The results revealed the important role that both dust and water vapour play in the heating rates within the Summer- and the Winter-SAL. The dominant dust-induced SW radiative warming (up to 0.7 K day$^{-1}$) is found slightly below the dust peak. However, we have confirmed that atmospheric water vapour has a

dominant radiative impact on the net SW warming observed within the Summer-SAL (from 2.1 km to 5.7 km) in addition to the strong cold anomaly near the SAL's top (-0.6 K day$^{-1}$) (LW adiabatic cooling). The higher water vapour content found to be carried on the Summer-SAL ($\sim 2$ g kg$^{-1}$) is very low but represents a high relative variation in comparison to the very dry free troposphere under clean conditions ($\Delta r \sim +332$ % at 5.3 km, roughly coinciding to the altitude of the maximum cooling). These results confirm the important radiative role played by water vapour on the atmospheric column, both at higher

levels near the STI and at lower levels counteracting the warming effect of aerosols in the SW. This is a relevant aspect to be well-thought-out in atmospheric modelling since ignoring the radiative effect of water vapour would overestimate the aerosol radiative effect. In the case of the Winter-SAL, the radiative effect has been attributed to the combined dust/moisture effect. This is the case of the SW heating found within the SAL (maximum near the peak observed in $r$), the dry anomaly at lower levels ($\Delta r \sim -38$ % at 1 km) and the thermal cooling ($\sim 0.3$ K day$^{-1}$) from the STI upwards. Thermal advection processes

were found to dominate in summer from 800 to 2.5 km height as a consequence of the strong temperature gradient between the ocean and the African continent in the lower troposphere. In contrast, radiative processes are found to dominate below 800 m in the MBL which result in heating, between 2.5 km and 6 km (heating), and above 6 km which produce cooling.

Finally, our results suggest the strong impact of the SAL in summer and winter on low clouds as well as the activation of heterogeneous ice nucleation processes under favourable moisture conditions in the presence of the Summer-SAL. As a





result, we have detected: 1) a lower occurrence of low clouds due to drier and warmer conditions at lower levels causing a strong suppressive influence on cumulus convection, with a more prominent effect in summer; and 2) a higher occurrence of mid-level and high-level clouds under SAL influence in summer. These mid- and high-level clouds are mostly located near the SAL's top, above the isozero level (between 5 km and 7 km height) at relatively warm temperatures (between -9°C and -15°C), and are likely to have a big impact on the surface energy budget. A case study of a 5-day dust event with MPL-3 data and

DREAM dust model INPC predictions was used to give some evidence of the role of dust and the water vapour carried within the SAL in ice nucleation processes. The convergence between lidar and dust model predictions has served us to verify, at least in the first instance, the impact of dust on cloud formation.

The results presented here introduce a robust characterisation of the Eastern Subtropical North Atlantic atmosphere and contribute to our understanding of the vertical/seasonal characteristics and the radiative impact of the SAL and its role in cloud

nucleation. This information is crucial to improve the current simulations and parameterizations of atmospheric processes in aerosol models. Our use of a long-time series of observations (12 years) concurrently carried out with independent and reference-type of measurements gives us confidence in our ability to closely characterise the SAL and the factors that impact its behaviour and the effects on the regional environment. However, we raise a number of issues that will require further studies to better assess the radiative effects of these modestly supercooled mid-level clouds as well as to incorporate the presence

of SAL-induced clouds in the evaluation of the radiative impact. Furthermore, given the high complexities inherent to ice nucleation processes, strongly dependent on $T$, $RH$ and dust concentration in a way not yet fully understood, an extended observation period should be used to elucidate the role of dust in ice initiation processes. All these analyses are out of the scope of this paper, being aimed to be carried out in future studies.

*Data availability.* The data from AERONET used in the present study can be freely obtained from https://aeronet.gsfc.nasa.gov (Holben et

al. 1998). The vertical soundings can be freely downloaded from http://weather.uwyo.edu/upperair/sounding.html. MOPSMAP simulations is a tool freely available at https://mopsmap.net/v1.0/mopsmap.php (Gasteiger and Wiegner, 2018). Lidar data used in this study can be provided by request to corresponding author Africa Barreto at abarretov@aemet.es.

## Appendix A: Abbreviations

| | |
|---|---|
| AOD | Aerosol Optical Depth |
| AE | Angstrom Exponent |
| AEMET | State Meteorological Agency of Spain |
| AERONET | Aerosol Robotic Network |
| ARTI | African air masses Residence Time Index |
| CALIOP | Cloud-Aerosol Lidar with Orthogonal Polarization |
| CCN | Cloud Condensation Nuclei |





| | |
|---|---|
| CBL | Convective Boundary Layer |
| CFT | Clean Free Troposphere |
| ECMWF | European Centre for Medium-Range Weather Forecasts |
| ERF | Effective Rariative Forcing |
| FLEXTRA | FLEXible TRAjectories model |
| FT | Free Trosposphere |
| GALION | GAW Aerosol Lidar Observation Network |
| GAW | Global Atmosphere Watch |
| GCCN | Giant Cloud Condensation Nuclei |
| GCOS | Global Climate Observing System |
| GPS | Global Positioning System |
| GUAN | GCOS Upper-Air Network |
| IARC | Izaña Atmospheric Research Center |
| INP | Ice Nucleation Particle |
| INPC | Ice Nucleation Particle Concentration |
| ITCZ | Inter-tropical Convergence Zone |
| IZO | Izaña Atmospheric Observatory |
| LR | Lidar Ratio |
| LW | Longwave |
| MBI | Marine Boundary Inversion |
| MBL | Marine Boundary Layer |
| MOPSMAP | Modeled optical properties of ensembles of aerosol particles |
| MPL | Micropulse Lidar |
| NAFDI | North African Dipole Intensity |
| NCAR | National Center for Atmospheric Research |
| NCEP | National Centers for Environmental Prediction |
| NMM | Non-hydrostatic Mesoscale Model |
| NRB | Normalized Relative Backscatter |
| OPAC | Optical Properties of Aerosols and Clouds |
| RH | Relative humidity |
| SAL | Saharan Air Layer |
| SCO | Santa Cruz station |
| SEM | Standard Error of the Mean |
| SHL | Saharan Heat Low |
| SNR | Signal-to-Noise Ratio |





| | |
|---|---|
| STI | SAL Temperature Inversion |
| SW | Shortwave |
| TI | Transition Inversion |
| TWI | Trade Wind Inversion |
| TWL | Trade Wind Layer |
| USGS | United States Geological Survey |
| WMO | World Meteorological Organization |

*Author contributions.* AB and EC designed the study and performed the analysis. RDG performed the radiative impact study. JC performed
the analysis based on thermodynamic soundings. LI, SB and SN were in charge of DREAM simulations. JP assisted with the interpretation
of the results. ABe and YH provided the results of the two-layer approach for lidar data. All authors discussed the idea and results of this
study.

*Competing interests.* The authors declare that they have no conflict of interest.

*Acknowledgements.* This work has been developed within the framework of the activities of the World Meteorological Organization (WMO)
Commission for Instruments and Methods of Observations (CIMO) Izaña Testbed for Aerosols and Water Vapour Remote Sensing Instru-
ments. AERONET sun photometers at Izaña have been calibrated within the AERONET Europe TNA, supported by the European Community
Research Infrastructure Action under the FP7 ACTRIS grant, agreement no. 262254. The LibRadtran Radiative Transfer Model has been
used to estimate the heating rates. The authors also want to acknowledges F. Molero for his useful tips on the atmospheric heating rates
retrieval and the funding provided by the Institute of Physics Belgrade, through the grant by the Ministry of Education, Science, and Tech-
nological Development of the Republic of Serbia. AEMET and BSC authors have participated in this study to contribute to the WMO Sand
and Dust Storm Warning Advisory System (SDS-WAS) Regional Center for Northern Africa, Middle East, and Europe.





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
