# Peer review of "Long-term characterisation of the vertical structure of the Saharan Air Layer over the Canary Islands using lidar and radiosondes profiles: implications for radiative and cloud processes over the subtropical Atlantic Ocean."

_Atmospheric Chemistry and Physics, 2021_

## Author Comment (AC2)

**The manuscript presents the seasonal evolution of meteorological vertical profiles and lidar-derived extinction coefficients using long-term observations in Tenerife. Data were analysed under the clean scenario, the summer-Saharan scenario, and the winter-Saharan scenario. Both dust and water vapour impacts in the radiative balance were investigated.**

**The ice nucleation processes were also discussed. The dataset is interesting, and the manuscript is well written. The manuscript is worthwhile to be published, after addressing all the points raised by reviewers.**

*Authors:*  *We acknowledge the referee's positive and constructive comments. Below we respond to his/her general comments.*

**Please see below some suggestions and comments:**

**The manuscript is long, and with too many acronyms. The Appendix A helps a bit, but I had to continuously go back and forth to remind myself what all those acronyms represent. Maybe reduce some used of abbreviations, e.g. clean FT (CFT). Besides, it would help if some important abbreviation definitions were added in the figure captions.**

This comment is similar to the referee's 1 technical comment #1.

*Authors:*  *Following these two suggestions, we have reduced the number of acronyms in the abstract. However, we consider the acronym MPL must be used since it is the common name of the most important instrument used in this study.*

[revised manuscript text omitted]

**In the abstract (l 8) and conclusion (l 600), "it was associated with lidar extinction coefficients ~ 0.030 km-1". Please clarify this "extinction coefficients" (max at which height? mean with standard deviation?). In line 265, you mentioned the maximum in the median extinction profile is ~0.033 km$^{-1}$.**

*Authors: We agree with this comment. The sentence is confusing. We propose to change the two sentences with these ones:*

> ***"A relatively well-mixed marine boundary layer (MBL) was observed in the case of clean (dust-free) conditions; it was associated with relatively constant lidar extinction coefficients ($\alpha$) below 0.036 km$^{-1}$ with minimum $\alpha$ (< 0.022 km$^{-1}$) in the free troposphere (FT)."***

> ***"The results of this study have revealed that, in the case of clean conditions, the MBL is relatively well-mixed with $\alpha$ values below 0.036 km$^{-1}$, and significantly clean conditions are observed in the free troposphere ($\alpha$ < 0.022 km$^{-1}$)."***

**l 9 add "+" for 48%**

*Authors: Done*

**l 10-11 clarify "lower levels" and "higher levels"**

*Authors: We agree with this referee comment. The information about the levels is included at the end of the sentence and can be confusing to the reader. We propose to reorder the sentence as follows:*

> *"The Summer-SAL appears as a well-stratified layer, relatively dry at lower levels **(Δr ∼−44 % at the SAL's base, where r is the water vapour mixing ratio)** but more humid at higher levels compared with clean FT conditions **(Δr ∼+332 % at 5.3 km)**, with a peak of α > 0.066 km−1 at ∼2.5 km."*

**l 128-131 this paragraph is more related to the section 2 "site"**

*Authors: We have moved this paragraph to the Section 2: Experiment Site.*

**l 154-158 The full overlap of MPL-3 is at ~ 5km agl. I assume you have applied overlap corrections in this study, was the same overlap correction applied to all profiles? The lower limit of the reliable backscatter / extinction profiles after the overlap correction is 300 m? With the uncertainty of only 10 % near ground? "a relative uncertainty in the overlap correction between 5 % and 10 %", do you mean uncertainty on the NRB or derived optical profiles?**

**Any vertical smoothing was applied in the profiles?**

*Authors: During the 12 year period with MPL-3 data a total of 6 overlaps have been performed at Izaña high altitude Observatory, ranging the full overlap altitude from 3.5 to 6.1 km. Different overlaps have been applied to the signal, considering the observed decay in the signal.*

*The lower limit of 300 m has been set taking into account both uncertainties due to overlap and afterpulse.*

*Regarding the uncertainty due to the overlap correction. Following Figure 5 in Welton and Campbell (2002), fractional percent uncertainty in the overlap correction is typically lower than 10%, from 3% to 6% between ground level to 6 km, where the full overlap is usually reached. However, following Sicard et al (2020), a non-appropriate overlap function applied in the retrieval of the backscatter coefficient yields errors up to 60% in the first 0.5 km and up to 20% above. We will update this information in the text:*

> *"Furthermore, the corrected profiles are hourly averaged to increase the signal-to-noise ratio (SNR). The MPL-3 installed at SCO station achieves a full overlap at an average of ∼5 km. Welton and Campbell (2002) estimated a relative uncertainty in the overlap **calculation typically below 10 %**. **Sicard et al (2020) estimated the errors of applying a non-appropriate overlap function to be up to 60% in the lowermost atmospheric layers and up to 20% above. Due to the overlap effect** in addition to the saturation of the*

*detector in the near range caused by the afterpulse phenomenon, the information below 300 m is disregarded."*

*We have not applied any vertical smoothing in the profiles.*

**l 159-167 LR were estimated using two-layer approach, and then applies to derive extinction coef. What were the LR values? These are interesting information to present.**

*Authors: We agree that LR is an important information to be included in the text, since these values have been used to derive the extinction coefficients. We propose to include the following information in the text:*

> *Sect. 4.1: "... We used a total of 10658 lidar profiles corresponding to 610 days in the case of the extended spring scenario, 5341 (217 days) for summer, 3842 (240 days) for autumn, and 1532 (136 days) for winter. **Average LR between 16 sr and 18 sr have been retrieved for the first layer, and between 48 sr and 50 sr for the second layer. These values are in agreement with previous studies performed by Berjón et al. (2019) in Tenerife with the same two-layer analysis.**"*

> *Sect. 4.2: "... We used a total of 3529 lidar vertical profiles (173 days) for summer, and 2437 lidar profiles (corresponding to 105 days) for winter. The main features of each dust scenario are shown in both Fig. 2 and Table 2. **Average LR of 19 sr (summer) and 15 sr (winter) have been retrieved for the first layer, and 47 sr (summer) and 51 sr (winter) for the second layer. These LRs are in agreement with the analysis performed by Berjón et al. (2019)."***

**What is the error/uncertainty on the derived extinction coefficients? Can it be added in the median extinction profile (e.g. by error bar)? I assume in the overlap region (below 5km), this uncertainty has higher value.**

*Following Bösenberg and Hoff (2007), typical relative errors in the retrieved backscatter and extinction profiles of 10 % and 20 % are assumed for the combined elastic lidar–photometer technique, which is low enough for climate impact studies. However, as many authors have shown (e.g. Kovalev, 1995; Barnaba and Gobbi, 2001; Pelón et al., 2002; Ansmann, 2006), considerably higher errors may be expected in the case of complex aerosol distributions such as under the presence of different aerosol layers in the vertical or horizontal inhomogeneous aerosol layers. Extinction coefficients have been retrieved in this paper following the two-layer approach published by Berjón et al. (2019). This methodology, an inversion method based on the Fernald–Klett method, combines Micropulse Lidar and photometric information considering two layers of aerosol with two different lidar ratios, one in the MBL and another one in the FT. This method is expected to better match the real lower-troposphere vertical structure of the North Atlantic subtropical region, commonly affected by the presence of a complex vertical layering because of the transport of Saharan dust. Since no specific uncertainty analysis was developed in the mentioned article, and considering this calculation is out of the scope of the present study, it is plausible to estimate the maximum uncertainty in the extinction retrieval at 20%, the highest typical error of the common inversion techniques.*

*These errors have been estimated without considering overlap. They are expected to be higher when the rest of the source of errors are included, such as the incomplete overlap, which is expected to be more important below the full overlap region.*

**Fig.1 Mean extinction profile is not in the grey shaded area, especially in fig.1b. Do you have an explanation?**

*Authors: It is true that, in the clean scenario at summertime above 4 km, mean values exceed the 20-80 percentile area. The authors attribute this departure to the presence of outliers in the extinction dataset at these altitudes, probably due to residual dust recirculated from previous dust outbreaks. In these conditions, ARTI=0 discards that the origin of the air mass in the previous 5 days is over Africa. However, given the high frequency of dust outbreaks in summer time, and considering the low-efficiency wet deposition processes to remove aerosols from the atmosphere in summer, residence time of dust injected above the MBL is considerably higher than in other seasons.*

**Please specify "lidar total extinction at each level".**

*Authors: We have added this information in the caption.*

**Is the height asl in all figures?**

*Authors: Yes. We have added this information in the caption.*

**Table2, for winter "and summer" seasons.**

*Authors: Done*

**Fig. 4,5 add "523nm" in the caption for extinction.**

*Authors: Done*

**l 547 fig.6b not a**

*Authors: Done, it was a typo.*

**l 563-564, information was not shown in fig.7b, add the reference of Fig S8 here. "-" are not visible in fig.7b, change the figure as in Fig S8.**

*Authors: We have modified this sentence as follows:*

> *"...The model predicts that most of the dust mass is in a layer between 2 and 6 km, in agreement with the MPL-3. Maximum concentrations (>100 µg/m$^3$) were reached on the afternoon of 20 August (see Fig. **S8**)..."*

**Fig.7 change "nINP"**

*Authors: Done*

**l567 m$^3$**

*Authors: Done, it was a typo.*

**fig S7, 1h time averaged MPL-3 profile centred on 10:54? please clarify.**

**add the time for the DREAM extinction profile. What's the shaded area for DREAM profiles?**

*Authors: MPL-3 profile was retrieved at 10:54. However, DREAM outputs are recorded every 3h, and therefore +3h (15 UTC) and -3h (9 UTC) DREAM profiles (centered around 12 UTC) are used to determine the shaded area to represent variability of the model results around measurement time. These profiles are bilinearly interpolated to the observation station latitude and longitude, because the model resolution is 0.1\*0.1 degrees.*

*We have added this information in the figure caption.*

---

## Author Response (AR2)

Manuscript: acp-2021-508
Title: Long-term characterisation of the vertical structure of Saharan dust outbreaks over the Canary Islands using lidar and radiosondes profiles: implications for radiative and cloud processes over the subtropical Atlantic Ocean.

**(1) Comments from referees/public & Author's response**

*__Anonymous Referee #1__*

**GENERAL COMMENTS**

The study by Barreto et al. focuses on the vertical profiles of aerosol extinction and meteorological variables over Tenerife during Saharan dust outbreaks, compared to clean scenarios. The analysis is based on 12 years of data and is differentiated between winter and summer. The dust layer and the marine boundary layer characteristics are studied separately. The direct radiative effects of both aerosol and water vapour advected during the outbreaks is assessed in both the short- and long-wave ranges based on average profiles. Finally, some considerations about ice nucleation processes are given based on cloud statistics and model results.

Although the paper is quite long and not always a compelling read, it condenses a great amount of information that the interested reader can have all in one place. Hence, the paper deserves publication on ACP. Some marginal comments follow, mainly based on the reviewer's expertise.

*Authors: We appreciate these positive comments. Specific comments are addressed below.*

**SPECIFIC COMMENTS**

**1. Is the asymmetry factor the best way of representing highly forward-scattering particles? Have the authors checked whether there is any difference if the full phase function is used?**

*Authors: The authors find this referee's comment interesting. The aerosol phase function is a key parameter to provide information on the angular intensity distribution of the scattered light. This is particularly true in the case of mineral dust, nonspherical particles (with significantly different backscattering properties in relation to spherical particles) with a characterized high forward scattering. However, the asymmetry factor, as an integral property of the phase function, is considered a computationally efficient parameter to replace the phase function in the study of aerosol radiative transfer properties, reducing significantly the efforts to parameterise the phase function (Horvath et al, 2015). As Boucher (1998) stated, parameterizations as the Henyey–Greenstein (HG) phase function, the one used in this study, can advantageously replace the phase function in most flux calculations with a small error, being generally considered acceptable for flux calculations. Following this author, HG phase function slightly overestimates dust forward scattering, and therefore appropriate corrections dependent on the aerosol size distribution, aerosol refractive index, and solar zenith angle should be included to refine the results.*

*In this paper we have used the scattering properties from MOPSMAP to model the radiative effect of dust, which uses spheres, spheroids, and a small set of irregular particle shapes over a wide range of sizes and refractive indices. Therefore, natural nonspherical aerosol particles, such as desert dust particles, are expected to be correctly modelled in terms of particle backscattering,*

*minimizing the potential errors associated with the effect of the particle shape on the aerosol direct radiative effect calculations. However, additional corrections related to the use of the HG phase function should ideally be included in future studies. The authors have not checked possible differences between full phase and modelled phase functions, but some detailed calculations can be extracted from previous publications (i.e., see Boucher 1998). We have added this information in the Sect. 3.5 of the manuscript:*

*"Atmospheric heating rate calculations are performed using the LibRadtran radiative transfer model (Mayer and Kylling, 2005; Emde et al., 2016). The simulations are done with the radiative transfer solver DISORT (DIScrete Ordinate Radiative Transfer) (Stamnes et al., 1988) for spectral ranges of 280 to 4000 nm (shortwave; SW) and 4000 nm and 100 μm (longwave; LW). The absorption parameterization REPTRAN (Gasteiger et al., 2014) included in the LibRadtran is used in both spectral ranges. Regarding the atmosphere gas composition, we use the standard midlatitude summer atmosphere model (Anderson et al., 1986). The temperature and relative humidity profiles are taken from the monthly or seasonally median profiles obtained from radiosondes (Sect. 3.3). **Henyey-Greenstein parameterization is assumed for computation of the aerosol phase function using the asymmetry parameter g as an approximation of the full phase function. Small errors but not negligible are expected from this approximation, generally considered acceptable for flux calculations (Boucher et al., 1998).**"*

**2. Have the authors compared the MOPSMAP output to the AERONET-derived aerosol properties within the common wavelength range?**

*Authors: Following the recommendation of the referee we have compared the AERONET-derived asymmetry factor values for dust in summer conditions (99 days) with the values extracted from AERONET inversions. The results are shown in the following table for the nearest common wavelengths, showing an excellent agreement between the two datasets under influence of mineral dust, which is the main objective of this study.*

|        | g AERONET | g MOPSMAP |
|--------|-----------|-----------|
| 440 nm | 0.748     | 0.757     |
| 500 nm | 0.730     | 0.744     |
| 675 nm | 0.742     | 0.715     |
| 870 nm | 0.747     | 0.706     |

**3. As mentioned at line 579, low cumulus clouds prevent observation of higher clouds using the lidar. Could the anticorrelation seen in Fig. 6 between low and high clouds be partly due to this limitation? In the specific case study, an all-sky camera was used to prove the presence of cirrus (not seen by the MPL-3). However, the cloud statistics in Fig.6 only rely on the lidar profiles.**

*Authors: The sentence "DREAM also predicts INPC on August 22 (from noon time) and 23 but the low cumulus clouds prevent observations of mid-level and cirrus clouds" (lines 578 and 579) refers to a particular case study in two specific days. In reality, with very few exceptions, both the stratocumulus layer and the altostratus/altocumulus layers (middle clouds) do not present a compact appearance. On the contrary, they are usually in the form of scattered clouds, a circumstance that allows the numerous lidar measurements carried out during daytime to detect not only low clouds but also medium and high clouds.*

*Given the relevance that these results have on the conceptual model of cloud-types distribution under clean and dust scenarios, and in the discussion of Sect. 7, we consider it appropriate to*

*introduce a second approach (included in the supplement) for the determination of the frequency of different types of clouds, totally independent of the results obtained using only MPL data.*

*We also propose to include a new section to clarify the cloud screening calculation with MPL-3. This new section 3.6 is the following:*

*3.6. Methodology for obtaining cloud type distribution for dust and clean scenarios*

*To determine the distribution of cloud types (low, middle and high clouds) for each clean and dust scenario in the 2007-2018 time period, we have estimated the cloud height using lidar data. Since the MPL system used in this study cannot measure particle depolarization, we are not able to directly estimate the presence of clouds and their respective altitudes. We addressed this issue by means of an indirect technique. We first extracted extinction coefficient values assuming a lidar ratio of 20 sr **in the classical Fernald-Klett inversion procedure,** close to the expected value for mid-level clouds (Yorks et al., 2011). Then, we have identified molecular (particle-free and therefore with the sole contribution of molecules) and aerosol signals **taking into account visibility and empirical thresholds in the value of the retrieved α and its quantile.** The presence of clouds is considered when **an empirical threshold in α is exceeded and** the visibility is reduced to values < 5 km if the molecular layer starts from the surface level (WMO, 2019). **Cloud is also identified when visibility is reduced to values < 10 km if the molecular layer starts above the surface level and empirical thresholds in α and in the 80% quantile of α are exceeded. "***

*In Sect. 7, the information included in the new Sect. 3.6 has been deleted. The text from line 507 to line 538 and Figure 6 have been replaced by the following information:*

*Lines 507 y 525: As we have shown in Sects. 5 and 6, both SAL's dust and humidity anomalies profoundly change the vertical radiative fluxes. **The relative frequency of cloud types from MPL data for both Saharan and clean scenarios, for winter and summer, has been calculated according to Sect. 3.6 in order to explore the effect of SAL radiative impact on cloud formation. The cloud type distribution is shown in Figures 6a and 6b, for winter and summer, respectively.** Analogous plots but from cloud climatological and total sky camera observations at 13 UTC are shown in Figures S6 of the supplementary material. Notice that these two datasets of direct and independent observations from two strategically located meteorological stations on the island of Tenerife (Tenerife North airport meteorological station at 600 m a.s.l. and Izaña meteorological station at 2400 m a.s.l.) are quite consistent with the lidar analysis. A notable reduction of low clouds in the case of the Saharan scenario (winter and summer) in comparison to clean conditions is observed using MPL data (Fig. 6 a and b). On the contrary, a higher occurrence of mid- and high level clouds under the Summer-SAL scenario are observed in Fig. 6 b (23 % under Saharan conditions versus 4 % in clean conditions). This pattern is corroborated, at least qualitatively, using direct cloud observations (Fig. S6). Although the reduction in low-level clouds is observed for both Winter-SAL and Summer-SAL, the reduction is more prominent in summer, when the probability of occurrence of low clouds is reduced from 40 % in summer clean conditions to 5 % under SAL conditions (Fig. 6b). In winter, this drop in low clouds is less pronounced, with a reduction from 25 % (winter clean conditions) to ∼ 4 % (Winter-SAL conditions).*

*Lines 538-540: **We can interpret some of the previous results in terms of the impact of dust on heterogeneous nucleation and the role of dust as ice nucleation particles (INP) (Weinzierl et al., 2017)**.*

*New Fig. S6 including subfigures a and b with relative frequency of clouds from the Oktas analysis for Tenerife Norte and Izaña, for winter and summer, respectively.*

[Figure]

*Fig S6. Frequency of occurrence (%) of low and middle/high clouds from the oktas analysis for Tenerife Norte Airport and Izaña Observatory for (a) winter and (b) summer.*

*New Sect. S5 in Supplementary material:*

*S5. Cloud observations from direct observations*

*In the classical approach, cloud observations in the time period 2007-2018 from two strategically located meteorological stations on the island of Tenerife have been used: the meteorological station at Tenerife North airport (WMO 60015) at 600 meters above sea level, excellent location to record low clouds (stratocumulus), and the Izaña meteorological station (WMO 60010) at 2400 m above sea level, permanently free of low clouds and therefore a very good site to observe middle (generally altostratus and altocumulus) and high clouds (cirrus). These two stations are 8 and 31 km away, respectively, from SCO. Climatological and routine observations performed at 13 UTC from Tenerife North airport provide the determination of oktas from trained staff, while an automatic determination of oktas around 13 UTC with a total sky camera was performed at Izaña. Using these two pieces of information we ensure that we can observe medium and high clouds when the low clouds do not allow us to see the first ones from the Santa Cruz station (SCO), at sea level, where the MPL is located.* ”

*TECHNICAL REMARKS*

**- abstract: acronyms should be avoided in an abstract unless a term is used multiple times (e.g., MPL-3 is only used one and other abbreviations only twice)**

*Authors: Following the referee's suggestions, we have reduced the number of acronyms in the abstract. However, we consider the acronym MPL must be used, since it is the common name of the most important instrument used in this study.*

*"Every year, large-scale African dust outbreaks frequently pass over the Canary Islands (Spain). Here we describe the seasonal evolution of atmospheric aerosol extinction and meteorological vertical profiles at Tenerife over the period 2007 – 2018 using long-term Micropulse Lidar (MPL-3) and radiosondes observations. These measurements are used to categorise the different patterns of dust transport over the subtropical North Atlantic and, for the first time, to robustly describe the dust vertical distribution in the Saharan Air Layer (SAL) over this region. Three atmospheric scenarios dominate the aerosol climatology: dust-free (clean) conditions, the summer-Saharan scenario (Summer-SAL) and the winter-Saharan scenario (Winter-SAL).*

*A relatively well-mixed marine boundary layer (MBL) was observed in the case of clean (dust-free) conditions; it was associated with relatively constant lidar extinction*

*coefficients (α) below 0.036 km$^{-1}$ with minimum α (< 0.022 km$^{-1}$) in the free troposphere (FT).* *The Summer-SAL has been characterised as a dust-laden layer strongly affecting both the MBL ( = +48 % relative to clean 10 conditions) and the free troposphere. The Summer-SAL appears as a well-stratified layer, relatively dry at lower levels but more humid at higher levels compared with* ***clean FT conditions*** *(r -44 % at the SAL's base and r +332 % at 5.3 km, where r is the water vapour mixing ratio), with a peak of > 0.066 km$^{-1}$ at 2.5 km. Desert dust is present up to 6.0 km, the SAL top based on the altitude of SAL's temperature inversion. In the Winter-SAL scenario, the dust layer is confined to lower levels, below 2 km altitude. This layer is characterised by a dry anomaly at lower levels (r -38 % in 15 comparison to the clean scenario) and a dust peak at 1.3 km height.* ***FT clean*** *conditions were found above 2.3 km.*

*Our results reveal the important role that both dust and water vapour play in the radiative balance within the Summer- and Winter-SAL. The dominant dust-induced shortwave (SW) radiative warming in summer (heating rates up to +0.7 K day$^{-1}$) is found slightly below the dust maximum. However, the dominant contribution of water vapour was observed as a net SW warming observed within the SAL (from 2.1 km to 5.7 km) and as a strong cold anomaly near the SAL's top (-0.6 K day$^{-1}$). The higher water vapour content found to be carried on the Summer-SAL, despite being very low, represents a high relative variation in comparison to the very dry clean free troposphere in the subtropics. This relevant aspect should be properly taken into account in atmospheric modelling processes. In the case of the Winter-SAL, we observed a dust-induced radiative effect dominated by SW heating (maximum heating of +0.7 K day$^{-1}$ at 1.5 km, near the dust peak); both dust and atmospheric water vapour impact heating in the atmospheric column. This is the case of the SW heating within the SAL (maximum near the r peak), the dry anomaly at lower levels (r -38 % at 1 km) and the thermal cooling (0.3 K day$^{-1}$) from the* ***temperature inversion*** *upwards.*

*Finally, we hypothesise that the SAL can impact heterogeneous ice nucleation processes through the frequent occurrence of mid-level clouds observed near the SAL top at relatively warm temperatures. A dust event that affected Tenerife in August 2015 is simulated using the regional DREAM model to assess the role of dust and water vapour carried within SAL in the ice nucleation processes. The modelling results reproduce the arrival of the dust plume and its extension over the island and confirm the observed relationship between the Summer-SAL conditions and the formation of mid- and high-level clouds.*"

**- l. 54: "studies... study" ("investigation" or "research" can be used instead?)**

*Authors: Yes, we have modified this text as follows:*

> *"Another problematic point of the current* ***research*** *is related to the preferential study of the dust leaving Africa at tropical latitudes in the summer season when the Saharan outbreaks over the North Atlantic are mostly**..*"*

**- l. 61-69: The Introduction is very long. Maybe these lines with detailed considerations might be dropped?**

*Authors: We agree with this comment, these lines have been deleted from the final manuscript.*

**- l. 78: "Therefore" is used as a consequence of what? It is not obvious that biomass burning aerosol has specific geographical provenance**

*Authors: We have used the word "Therefore" in the text to express consequence. Based on our experience, dusty air masses getting the Canary Islands in winter are commonly caused by deep lows or cut-off lows that became detached from the mid-latitudes circulation located in the vicinity of the Canary Islands archipelago transporting mineral dust within a relatively short distance (see Cuevas et al. 2021 and references therein). This fast and short-distance transport is commonly originated in the Western Sahara, a desert region with very sparse vegetation, and therefore with a very low probability of being a source of biomass burning aerosols, as confirmed by our analysis of dust composition. This situation contrasts to dust transport in winter in the tropics, often mixed with species linked to biomass burning from the Sahel region during the dry season (Formenti et al., 2003; Capes et al., 2008; Rodríguez et al., 2011).*

**- l. 98: "from Tenerife" (Tenerife was already mentioned one line above) could maybe replaced by "in the same area"?**

*Authors: Done*

**- l. 161-163: information on how aerosol properties at slightly different wavelengths are compared (mentioned at lines 441-443) could be anticipated here**

*Authors: The information about MOPSMAP provided in lines 441-443 is used to compute diurnal-averaged heating rates. However, in lines 161-163, we describe the methodology to retrieve aerosol extinction profiles from MPL-3 data. The authors think these two are independent pieces of information that have been presented in different sections as it has done in the manuscript.*

**- l. 183: only arrival altitudes are mentioned. Does ARTI also accounts for the trajectory altitude above the Saharan-Sahel surface?**

*Authors: ARTI index is computed using 120-hour back-trajectories arriving at two levels, 150 m and 2400 m, which are considered the most representative levels to study air mass long transport arriving at SCO and IZO, respectively. This index expresses the percentage of the time in which the air-mass trajectory is above the Saharan-Sahel surface, so that 0% means that a specific air mass has not passed above the Saharan-Sahel region, and high values of ARTI indicate an African origin of the air mass getting SCO or IZO. So, the answer to the referee is yes, the ARTI index accounts for the trajectory of the air masses above the Sahara-Sahel surface.*

**- l. 189: it could be anticipated that an additional criterion based on aerosol quantities**

**derived by AERONET (l. 247-250) is also used to identify the dust air masses**

*Authors: We use in this paper two independent pieces of information to discriminate between clean and dust conditions, described separately in Sect. 3.1 (AERONET) and 3.4 (ARTI). ARTI ensures clean conditions to be met (ARTI=0). This restriction, although very restrictive, ensures that we correctly select clean days. The other one is the AOD and AE extracted from AERONET, to distinguish between clean and dust conditions. The authors consider that anticipating this information in Sect. 3.4 is not enlightening but can add confusion to the reader.*

**- l. 228: this sentence is too general and unclear. Please, rephrase**

*Authors: We have clarified the sentence:*

> ***"Atmospheric scenarios have been classified using two different and independent techniques: lidar extinction profiles, able to characterize the vertical distribution of aerosols, and atmospheric soundings, necessary to characterize the atmosphere from***

*a thermodynamic perspective. Only daytime lidar profiles and atmospheric soundings (launched at 12 UTC) have been included in this study."*

**- l. 267: is the presence of residual dust a sign that 120h is a too short period for the backtrajectory calculations?**

*Authors: The presence of some residual dust is attributed to the maximum extinction values only sporadically observed in clean conditions during winter. Dust arrival is confined to lower levels this time of the year, as is shown in the paper. The value of 120h was selected as a compromise between reliably detecting the origin of the air mass and limiting the increasing uncertainty as we go further back in time. The authors consider that, despite this residual dust observed in the winter clean scenario might be minimized using a longer time period, it would be counterproductive in terms of geographical uncertainty.*

**- l. 267-268: the two sentences are maybe a bit confusing? It sound like both the second extinction maximum and the minimum are located at the same place, i.e. near the MBL top**

*Authors: The authors agree with this comment. We have changed this sentence.*

**"Mean and median lidar profiles (Fig. 1) show a rather constant α profile within the MBL, with two secondary maximum values. The first maximum in the median profile, with α ~ 0.033 km⁻¹, is located at altitudes ranging from 0.6 km in summer to 0.7 km in autumn-winter. Maximum α peak values are found in winter (0.036 km⁻¹), which may be a consequence of the residual dust due to the frequent dust outbreaks at this level in this season. The second maximum (α ~ 0.023 km⁻¹) is located at ~ 1.5 -1.7 km height. Minimum α values within the MBL are observed from 1.6 km in summer to 1.3 km in the other seasons. "**

**- l. 297: what does "500 km above Western Sahara" mean here? Is it "north of"? Same at line 358**

*Authors: Yes, this sentence is confusing. We propose to change these two sentences with these new ones:*

*"The temperature gradient in summer in the lower troposphere (e.g. 925 hPa) between Tenerife **and a point located 500 km away, in the Western Sahara,** is roughly 10 K ..."*

*"In this season, the temperature difference in the lower troposphere (e.g. 925 hPa) between Tenerife **and a point located 500 km away, in the Western Sahara,** is barely 2 K..."*

**- l. 374: please, state the altitude levels used to estimate the temperature inversion**

*Authors: The calculation of the height and the temperature of the inversion is performed individually (not at averaged levels) in each one of the atmospheric soundings. First, it is detected if there is inversion, when the temperature gradient acquires positive values, that is, the temperature increases with the height, according to the following equation (Carrillo et al., 2016):*

$$\Gamma i, j ((z2 + z1)/2) = \frac{\Delta a, i, j}{\Delta zi, j} = \frac{aj - ai}{zj - zi}$$

Where $\Gamma_{i,j}$ is the lapse rate of the "a" parameter between the $z_i$ and $z_j$ altitude level. The location of inversion layers where temperature lapse rate acquires positive values was considered. If there is an inversion, the base is determined as that point at which the temperature begins to increase with altitude, and the top, where it begins to decrease. At these points (base, top): height, temperature, etc. are determined. Note that the soundings are previously interpolated linearly every 10 hPa, and that the significant points are collected in the meteorological soundings. See Carrillo et al. (2016) for further details on the temperature inversion determination.

**- l. 399-400: "As a consequence" does not refer to the previous sentence, but to the one before. Please, reorder or rephrase**

*Authors: We agree with this comment. The paragraph has been reworded as follows:*

> *"These r anomalies, although small as absolute values, represent a high relative variation in comparison to the extremely dry FT under clean conditions (Δr up to +332 % at 5.3 km, roughly coinciding with the altitude of the maximum cooling). As a consequence, the SAL appears as a relatively moist layer at mid-levels. The anomaly in α peaks at 2.6 km, where maximum extinction values are commonly retrieved (Fig. 3 (c))."*

**- l. 427: acts "in" modifying?**

*Authors: We have corrected the mistake as follows:*

> *" … this dust-laden layer acts **by** significantly modifying the thermodynamic profiles…"*

**- l. 514-519: have the authors validated the cloud screening procedure described, and how? How different are thick Saharan dust layers from clouds, as seen by the MPL?**

*Authors: The authors have included more information in the manuscript to ensure the validity of this cloud screening. Independent observations of clouds have served as validation, as is written in the new Sect. S5 of the supplementary material (see specific comment #3).*

*Regarding the thickness of the Saharan dust layers, as seen by the MPL, with a vertical resolution of 75m, the MPL (using the aforementioned cloud screening procedure) is able to detect if a specified bin (layer) is affected by the molecular (Rayleigh) regime, or is affected by aerosols or clouds. The difference between aerosol and cloud is expected to be different from the lidar signal. We have based these differences on empirical thresholds determined in the extinction values (FK inversion assuming LR=20 sr), also dependent on visibility. If visibility is lower than 10 km (according to WMO), clouds are identified if the 80% quantile is higher than 0.3912 $km^{-1}$ or relative humidity is higher than 70%, or extinction is higher than 0.3912. If visibility is lower than 5 km and the layer starts from the surface level, a cloud is identified if extinction is higher than 0.7824 $km^{-1}$.*

*We have added a new Sect. 3.6 in the text and more information in the text to clarify this classification (see comment #3).*

**- Fig. 10: "ln10" --> "log10"**

*Authors: Done*

*References:*

*Horvath, H.: Extrapolation of a truncated aerosol volume scattering function to the far forward and back region, J. Aerosol Sci., https://doi.org/10.1016/j.jaerosci.2015.08.001, 2015.*

*Boucher, O. (1998). On Aerosol Direct Shortwave Forcing and the Henyey–Greenstein Phase Function, Journal of the Atmospheric Sciences, 55(1), 128-134. Retrieved Oct 21, 2021*

**(2) Author's changes in the manuscript**

Please, find the following changes performed by the authors after the referee's corrections/suggestions. Lines number are referred to the corrected manuscript.

Authors: We have corrected the affiliation of Luka Illic (Institute of Physics, University of Belgrade, Belgrade, Serbia).

Title: Taking into account the internal comments between authors, as well as suggestions from scientists external to this work, it is proposed to slightly change the title of the paper to include the clear and defined concept of "Saharan Air Layer" instead of "Saharan dust outbreaks". Authors consider that this title better defines the object of study and agrees better with the definitions of the different scenarios (Winter-SAL and Summer-SAL). So, the new title is "Long-term characterisation of the vertical structure of **the Saharan Air Layer over** the Canary Islands using lidar and radiosondes profiles: implications for radiative and cloud processes over the subtropical Atlantic Ocean".

L8: Changed "relatively constant" by "**rather** constant".

L10: "Free troposphere" has been changed by "FT" to be consistent with the acronym previously defined.

L48: Changed "represents the major radiative driver for the dust layer in summer" by "represents the major radiative driver for the **SAL** in summer".

L91: Changed "We use concurrent sun photometer measures to minimize the uncertainties involved in the aerosol extinction retrieval from an elastic backscatter system" with "Sun photometer measures allow us to minimize the uncertainties involved in the aerosol extinction retrieval from an elastic backscatter system".

L95: Full stop here.

L121: Typo corrected: "Global Atmospher**e** Watch"

L140: Correct mathematical notation is "365 days per year"

L154: AODs were extracted at 523nm at IZO and SCO. We have corrected this mistake in the text.

L156: Changed "This iterative procedure enables the LR to be calculated for each layer, the boundary layer (principally containing marine and dust aerosols) and the lofted aerosols in the free troposphere" with "This iterative procedure enables the LR to be calculated for **the boundary layer** (principally containing marine and dust aerosols) and the lofted aerosols **layer** in the free troposphere".

L169: Changed "are measured using Vaisala RS92 radiosondes" with "**have been** measured using Vaisala RS92 radiosondes".

L181: Changed "to study the long transport arriving at SCO and IZO" with "to study the **long-range** transport arriving at SCO and IZO".

L193: Changed "Henyey-Greenstein parameterization is assumed for computation of the aerosol phase function" with "Henyey-Greenstein parameterization is assumed for **computing** the aerosol phase function".

L207: New information added and the sentence has been rephrased:

"To determine the distribution of cloud types (low, middle and high clouds) **and therefore to study the possible impact of the SAL on cloud formation** we have estimated the cloud height using lidar data for each clean and dust scenario in the 2007-2018 daytime period. **In this way, a dataset of 16400057 bins at different altitudes classified as cloudy corresponding to 2877 different days has been used in this cloud classification.**"

L246: Comma deleted.

L247: Deleted "ARTI (defined in Sect. 3.4 is helpful to define the clean scenarios.".

L249: Deleted "(see Suplementary material)"

L261: Added in formation in : "10658 lidar **cloud-free** profiles"

Table 2: Corrected a typo "and".

L264: Changed "These values are in agreement with previous studies performed by Berjon et al. (2019)…" with "These values are in agreement **with those obtained in a previous study** performed by Berjon et al. (2019)…".

L267: Deleted "A seasonal variability of the MBL thickness is observed, extending up to 1621 m in autumn and to 1238 m in summer.".

L271: Changed "It explains the greater vertical extension of the MBL in September-October because subsidence is at a minimum and its lower vertical extension in summer when subsidence is at a maximum." with "It explains the greater vertical extension of the MBL **observed** in September-October (**1238 m**) because subsidence is at a minimum and its lower vertical extension in summer (**1621 m**) when subsidence is at a maximum.".

L276: Changed "which may be a consequence of the residual dust due to the frequent dust outbreaks at this level in this season." with "which may be a consequence of residual **dust caused by** the frequent dust outbreaks at this level in this season.".

Figure 1 caption: Deleted "detected from vertical soundings".

L301: Changed "the dust intrusions do not constitute an elevated layer but a dust transport confined to lower levels" with "the dust intrusions do not constitute an elevated layer but a dust **layer** confined to lower levels".

L302: Added in formation in : "We used a total of 3529 lidar vertical **cloud-free** profiles (173 days) for summer, and 2437 lidar **cloud-free** profiles".

L305: Changed "These LRs are in agreement with the analysis performed by Berjón et al. (2019)." with "These LRs are **consistent with the LR values reported** by Berjón et al. (2019)".

L309: Proper reference to a figure in the Supplement: "The temperature gradient in the lower troposphere (e.g. 925 hPa) between Tenerife and a point located 500 km away, in the Western Sahara, is roughly 10 K or higher **(Fig. S5 (a))**".

L312: Deleted "lower boundary layer, the".

Figure 2 caption: They order in the use of the acronym SAL was incorrect "Orange shaded areas indicate the presence of the **Saharan Air Layer (SAL)**. Orange shaded areas with black spots in winter indicate a mixture of the **SAL** with the MBL".

L322: Changed "$\vartheta$ at surface level does not change its value" with "**At surface level, $\vartheta$** does not change its value".

L361: Deleted "wind".

L371: Proper reference to a figure in the Supplement: "In this season, the temperature difference in the lower troposphere (e.g. 925 hPa) between Tenerife and a point located 500 km away, in the Western Sahara, is barely 2 K **(Fig. S5 (b))**".

L413: Deleted "As a consequence, the SAL appears as a moist layer at mid-levels because this layer transports relatively moist air at altitudes where atmospheric humidity is commonly very low.".

L418: Changed "mixing ratio" with "r".

L436: Changed "As a result, both dust and dry anomalies play an important role in heating of the SAL in the lower troposphere (Wong at al., 2009)" with "As a result, both dust and dry anomalies play an important role in heating of the SAL in the lower troposphere **within the MBL** (Wong at al., 2009)".

L448: Added a reference to the section: "Simulations have been done including in LibRadtran spectral information of $\alpha$, SSA and g extracted from MOPSMAP (Gasteiger and Wiegner, 2018) (see Sect. 3.5)".

L457: Changed "summer clean" with "summer-clean".

L487: Added a comma after especially.

L515: Changed "so temperature advection processes do not take place" with "so temperature advection processes are not relevant".

L520-530: Rephrased as follows:

"As we have shown in Sects. 5 and 6, both SAL's dust and humidity anomalies profoundly change the vertical radiative fluxes. The relative frequency of cloud types from MPL data for both Saharan and clean scenarios, for winter and summer, has been calculated using the methodology described in Sect. 3.6, in order to explore the effect of SAL radiative impact on cloud formation. The cloud type distribution is shown in Figs. 6 (a) and (b), for winter and summer, respectively. Analogous plots but from cloud climatological and total sky camera observations at 13 UTC are shown in Fig. S6 of the supplementary material. A notable reduction of low clouds in the case of the Saharan scenario (winter and summer) in comparison to clean conditions is observed using MPL data (Fig. 6 (a) and (b)). On the contrary, a higher occurrence of mid- and high level clouds under the Summer-SAL scenario are observed in Fig. 6 (b) (23 % under Saharan conditions versus 4 % in clean conditions). This pattern is corroborated, at least

qualitatively, using direct and independent cloud observations (Fig. S6). Although the reduction in low-level clouds is observed for both Winter-SAL and Summer-SAL, the reduction is more prominent in summer, when the probability of occurrence of low clouds is reduced from 40 % in summer-clean conditions to 5 % under SAL conditions (Fig. 6 (b)). In winter, this drop in low clouds is less pronounced, with a reduction from 25 % (winter clean conditions) to ≈ 4 % (Winter-SAL conditions). This observed reduction is explained taking into account that the SAL transports dryer air at lower levels in comparison to the MBL's air humidity (both in summer and winter), and therefore both the lifting condensation level and the level of free convection rise, increasing the energetic barrier to convection".

L542: Changed "which can cause dust to be transported into the MBL due to entrainment at the base of the SAL much more rapidly than the transport caused by the gravitational settling" with "which can cause dust to be transported into the MBL due to entrainment at the base of the SAL much more rapidly than **dust vertical transport** caused by gravitational settling".

L548, 552, 554, 560, 561, 581, 582: Replaced "mid-level" with "middle".

L555: Replaced "after" with "above".

L562: Deleted "a case study of".

L563: Replaced "a dust event that impacted Tenerife in summer is simulated by the dust regional DREAM atmospheric model" with "a dust event that impacted Tenerife in summer **has been** simulated by the dust regional DREAM atmospheric model".

L571 and 574: Proper mathematical notation in $\mu g\ m^{-3}$

L585: Replaced "mid-level and cirrus clouds" with "middle and high clouds".

L590: Corrected "The convergence between lidar observations and dust model predictions allow**s** us to link".

L612: deleted "layer"

L629: Included "**cloud-free** lidar"

L642: 800 **m**

Caption Fig. 7: Corrected "**on** August 2015"

L601: Changed "Tenerife is a key location for Saharan dust studies" with "Tenerife is a key location **for** Saharan dust studies".

L602: Deleted "the".

L604: Changed "The study of the SAL in the North hemisphere subtropical region will add important additional information" with "The study of the SAL in the North hemisphere subtropical region **adds** important additional information".

L606: Replaced "and" with "while".

L612: Changed "Under SAL influence, the CBL reaches altitudes normally characterised by clean free troposphere conditions and Saharan air masses are more humid than FT at the same levels" with "Under SAL influence, the CBL reaches altitudes normally characterised by clean free troposphere conditions, **being** Saharan air masses more humid than FT at the same levels".

L622: Replace "Important" with "Significant".

L648: Replaced "mid-level and high-level clouds" with "middle and high clouds".

Appendix A: Deleted abbreviations of ITCZ, NAFDI and SHL not used in the text.

References added:

*Boucher, O. (1998). On Aerosol Direct Shortwave Forcing and the Henyey–Greenstein Phase Function, Journal of the Atmospheric Sciences, 55(1), 128-134. Retrieved Oct 21, 2021.*

Sicard, M.; Rodríguez-Gómez, A.; Comerón, A.; Muñoz-Porcar, C. Calculation of the Overlap Function and Associated Error of an Elastic Lidar or a Ceilometer: Cross-Comparison with a Cooperative Overlap-Corrected System. Sensors 2020, 20, 6312. https://doi.org/10.3390/s20216312.

References deleted as a consequence of referees's comments:

Cuevas, E., Gómez-Peláez, A., Rodríguez, S., Terradellas, E., Basart, S., García, R., García, O., and Alonso-Pérez, S.: The pulsating natureof large-scale Saharan dust transport as a result of interplays between mid-latitude Rossby waves and the North African Dipole Intensity,Atmos. Environ., 167, 586 – 602, https://doi.org/10.1016/j.atmosenv.2017.08.059, 2017a.

Lavaysse, C., Flamant, C., Janicot, S., Parker, D., Lafore, J.-P., Sultan, B., and Pelon, J.: Seasonal evolution of the West African Heat Low:880A climatological perspective, Clim. Dyn., 33, 313–330, https://doi.org/10.1007/s00382-009-0553-4, 2009.

Rodríguez, S., Cuevas, E., Prospero, J. M., Alastuey, A., Querol, X., López-Solano, J., García, M. I., and Alonso-Pérez, S.: Modulation ofSaharan dust export by the North African dipole, Atmos. Chem. Phys., 15, 7471–7486, https://doi.org/10.5194/acp-15-7471-2015, 2015.

Tegen, I., Schepanski, K., and Heinold, B.: Comparing two years of Saharan dust source activation obtained by regional modelling andsatellite observations, Atmos. Chem. Phys., 13, 2381–2390, https://doi.org/10.5194/acp-13-2381-2013, 2013.